# An anti-influenza A virus microbial metabolite acts by degrading viral endonuclease PA

Jianyuan Zhao[1,5], Jing Wang[1,5], Xu Pang[1,5], Zhenlong Liu[2], Quanjie Li[1], Dongrong Yi[1], Yongxin Zhang[1], Xiaomei Fang[1], Tao Zhang [1], Rui Zhou[1], Tao Zhang[1], Zhe Guo[1], Wancang Liu [1], Xiaoyu Li[1], Chen Liang[2], Tao Deng[3], Fei Guo[4✉], Liyan Yu[1✉] & Shan Cen [1✉]

The emergence of new highly pathogenic and drug-resistant influenza strains urges the development of novel therapeutics for influenza A virus (IAV). Here, we report the discovery of an anti-IAV microbial metabolite called APL-16-5 that was originally isolated from the plant endophytic fungus *Aspergillus* sp. CPCC 400735. APL-16-5 binds to both the E3 ligase TRIM25 and IAV polymerase subunit PA, leading to TRIM25 ubiquitination of PA and sub-sequent degradation of PA in the proteasome. This mode of action conforms to that of a proteolysis targeting chimera which employs the cellular ubiquitin-proteasome machinery to chemically induce the degradation of target proteins. Importantly, APL-16-5 potently inhibits IAV and protects mice from lethal IAV infection. Therefore, we have identified a natural microbial metabolite with potent in vivo anti-IAV activity and the potential of becoming a new IAV therapeutic. The antiviral mechanism of APL-16-5 opens the possibility of improving its anti-IAV potency and specificity by adjusting its affinity for TRIM25 and viral PA protein through medicinal chemistry.

[1] Institute of Medicinal Biotechnology, Chinese Academy of Medical Sciences and Peking Union Medical School, 100050 Beijing, PR China. [2] Lady Davis Institute for Medical Research, Jewish General Hospital, McGill University, Montreal H3T 1E2, Canada. [3] Institute of Microbiology, Chinese Academy of Sciences, 100101 Beijing, China. [4] Institute of Pathogen Biology, Chinese Academy of Medical Sciences and Peking Union Medical School, 100730 Beijing, PR China. [5] These authors contributed equally: Jianyuan Zhao, Jing Wang, Xu Pang. ✉email: guoafei@ipbcams.ac.cn; Liyanyu@imb.pumc.edu.cn; shancen@imb.pumc.edu.cn

The influenza virus is one of the major human pathogens that can cause respiratory disease. It is estimated that annual seasonal flu leads to 3–5 million cases of severe illness and up to 650,000 deaths[1]. Seasonal influenza vaccines do not replace anti-influenza drugs, especially in the event of unexpected influenza pandemics, or for the treatment of patients who respond poorly to vaccination[2]. Because of the emergence and transmission of influenza H3N2 and H5N1 strains, which are resistant to the M2 ion channel blockers amantadine and rimantadine[3], neuraminidase (NA) inhibitors oseltamivir and zanamivir which have been prescribed for treating most influenza virus infections[4,5]. Recently, the nucleoprotein inhibitor baloxavir marboxil was developed to target the cap-dependent endonuclease activity of the viral polymerase subunit PA[6,7]. However, approximately 90% of the influenza strains circulating during the 2008–2009 flu season were found to be resistant to NA inhibitors. The new inhibitor baloxavir marboxil presented a low genetic barrier against pre-existing resistance[8,9]. Therefore, novel and effective drugs are urgently needed to control influenza virus infection, particularly the emergence of new highly pathogenic influenza strains.

Viral enzymes have been the primary targets of the majority of the antiviral drugs approved for clinical use. In the same vein, the IAV RNA polymerase complex has long been pursued for the discovery of anti-IAV drugs. The functional IAV RNA polymerase complex contains viral proteins PA, PB1, PB2 and NP. The function of NP is to coat and protect viral RNA. PA is an endonuclease, cleaves in the 5' region of host mRNA. The 10 to 15 nt 5' mRNA fragment is anchored to the PB2 subunit and used by the PB1 polymerase as a primer to initiate the synthesis of nascent viral RNA[10]. This mechanism of viral RNA synthesis is named "cap snatching transcription". A few drugs have already been developed that target the IAV RNA polymerase complex[11]. The most successful one is balovaxir that inhibits PA endonuclease activity, and has been licensed in Japan and the United Sates. Pimodivir acts on PB2 and prevents PB2 from binding the 5' cap structure of cellular mRNA fragment. Favipiravir can inhibit PB1 polymerase activity.

Here, we report that a natural compound called APL-16-5 (previously named asperphenalenone E), which was isolated from the plant endophytic fungus *Aspergillus* sp. CPCC 400735[12], significantly inhibits the replication of influenza virus in cultured cells and mice. Our data showed that APL-16-5 acts by binding to both the E3 ligase tripartite motif containing 25 (TRIM25) and viral polymerase subunit PA. This allows TRIM25 to recognize and ubiquitinate PA which is subsequently degraded by proteasome. Our results support the development of APL-16-5 as a new anti-influenza drug in light of its protection of mice from lethal influenza virus infection.

## Results

### Identification of APL-16-5 as a potent anti-IAV drug targeting de novo viral replication.
During screening of microbial metabolites for IAV inhibitors, we identified several phenalenone derivatives, including APL-16-5 and APL-16-1 (Fig. 1a). These inhibitors exhibited potent anti-IAV activity, with submicromolar $EC_{50}$ ranging from 0.28 to 0.36 μM (Fig. 1b), determined by infecting a HEK293T-Gluc reporter cell line with influenza A virus A/WSN/33, as previously described[13]. In contrast, a phenalenone derivative APL-16-2 showed negligible inhibition of IAV, with an $EC_{50}$ of 61.2 μM (Fig. 1a, b). All phenalenone derivatives tested herein exhibited low cytotoxicity, with $CC_{50} > 100$ μM in several cell lines (Supplementary Fig. 1a), suggesting that the observed antiviral effect was not due to cytotoxicity. APL-16-5 presented the strongest IAV inhibition and was therefore selected for further investigation. Next, we found that in addition to inhibiting IAV in HEK293T cells, APL-16-5 potently

suppressed multiple strains of IAV and influenza B virus in both A549 and MDCK cells (Supplementary Table 1). In contrast, no significant inhibition of hepatitis C virus or ZIKV was observed (Supplementary Fig. 1b, c). Together, these data suggest that APL-16-5 is a potent and specific inhibitor of the influenza virus.

To investigate the anti-IAV mechanism by APL-16-5, a time-of-addition experiment was performed using a single-round IAV infection as previously described[13], in order to determine which step of IAV replication was inhibited by APL-16-5. The results showed that APL-16-5 effectively inhibited IAV infection when added within 2–6 h, particularly within 2–4 h post infection when IAV RNA is transcribed and inhibited by RNA polymerase inhibitor ribavirin (Fig. 1c). The relatively prolonged inhibition by APL-16-5 (up to 6 h) compared with ribavirin (up to 4 h) may result from the different antiviral mechanisms of these two drugs. We found no effect of APL-16-5 on the nuclear import of viral ribonucleoprotein (RNP) immediately after IAV infection, as opposed to the marked inhibition of viral RNA nuclear import by the known IAV entry inhibitor nucleozin[14] (Supplementary Fig. 1d), suggesting that APL-16-5 does not affect the early post-entry event until viral RNP enters the nucleus. Moreover, a similar inhibitory effect of APL-16-5 was observed in cells infected with a single-round IAV (Supplementary Fig. 1e), compared to that of wild-type IAV, suggesting that APL-16-5 exerts its antiviral activity prior to the late stage of viral infection and reinfection of progeny virus. These data suggest that APL-16-5 most likely inhibits viral RNA transcription.

To further examine this antiviral mechanism, the effect of APL-16-5 on viral RNA transcription was assessed using the IAV mini-genome replicon A549-5Ps, a cell line that stably expresses IAV RNA-dependent RNA polymerase (RdRp) subunits (PA, PB1, and PB2), NP, and a vRNA-like reporter gene *Gaussia luciferase* (*Gluc*)[13]. The results showed that APL-16-5, but not the control compound APL-16-2, caused a dose-dependent reduction in Gluc expression (Fig. 1d). Further supporting the impairment of viral RdRp function by APL-16-5, the levels of all three species of viral transcripts (vRNA, cRNA, and mRNA) were significantly reduced in IAV-infected cells treated with APL-16-5, but not in those treated with APL-16-2 (Fig. 1e and Supplementary Fig. 1f). These data suggest that APL-16-5 inhibits influenza virus infection by diminishing viral RNA levels.

### APL-16-5 induces proteasome-dependent degradation of PA.
Next, we investigated how APL-16-5 diminishes viral RNA levels. First, the effects of APL-16-5 on the levels of PB1, PB2, PA, and NP, which are expressed in the IAV mini-genome replicon were assessed. Interestingly, treatment with APL-16-5 (10 μM) markedly decreased the levels of all viral RNP components, while the PA protein was most sensitive to APL-16-5 treatment at a lower concentration of 2 μM (Fig. 2a). Since the expression of each viral gene in the IAV mini-genome replicon is controlled by the promoter in the plasmid DNA, independent of viral RdRp activity, the inhibitory effect observed with APL-16-5 did not result from defective RdRp. Thus, the RdRp inhibitor ribavirin, a nucleoside analog, inhibited RdRp activity (Fig. 1d), but did not affect the expression of viral RdRp subunits (Fig. 2a). To examine the response of each subunit of the viral RdRp complex to APL-16-5, we expressed PA, PB1, PB2, and NP individually. APL-16-5 was found to cause a significant reduction in the level of PA, but not in that of PB1, PB2, or NP (Fig. 2b), which was corroborated by quantification analysis of Western blot (Supplementary Fig. 2a). As a control, treatment with APL-16-2 had no effect on the expression of these viral proteins (Supplementary Fig. 2b). Previous studies have shown that co-expression of the RdRp subunits increased the level of each component compared with the

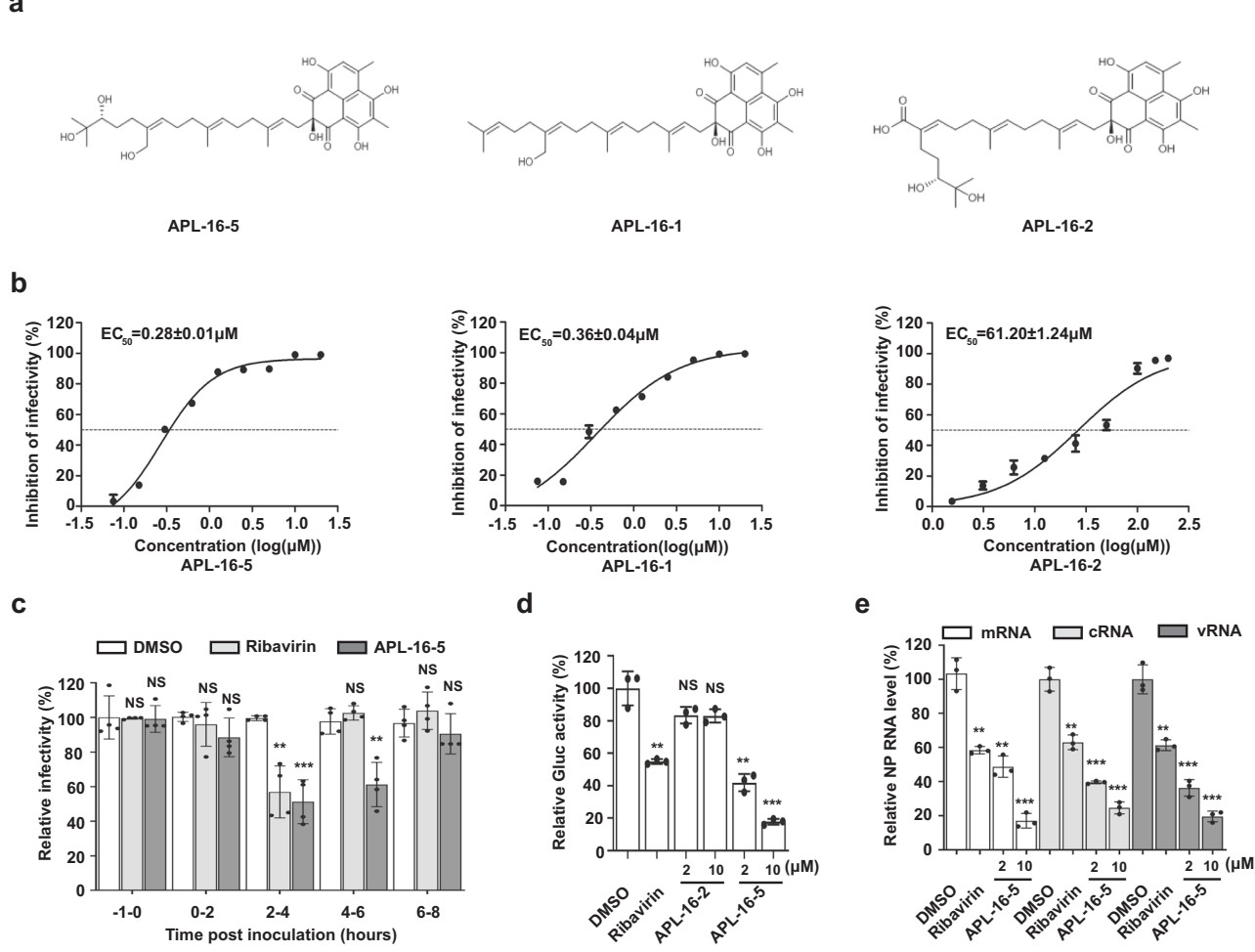

**Fig. 1 Identification of APL-16-5 as a potent IAV inhibitor targeting de novo viral replication. a** Molecular structure of compounds APL-16-5, APL-16-1 and APL-16-2. **b** Dose–response curves for the antiviral activity of APL-16-5, APL-16-1 and APL-16-2 were generated based on Gluc activity in HEK293T-Gluc cells infected with WSN/33 at a multiplicity of infection (MOI) of 0.5 for 24 h, then Gluc activity was determined ($n = 3$). Bar = mean. Error bars = ±SEM. **c** A time-of-addition experiment was performed to determine the step of IAV infection targeted by APL-16-5. A549 cells were infected with a single-round IAV at an MOI of 0.2. APL-16-5 (10 µM) was added before infection (−1 h), and at the indicated time of infection. At 11 h post-infection, Gluc activity was determined ($n = 4$). Ribavirin (20 µM) was used as a positive control. The drug was washed off at the end of the treatment period. Bar = mean. Error bars = ±SEM. For −1 to 0, 0–2, 2–4, 4–6, 6–8 h (DMSO vs. Ribavirin and APL-16-5): ($p = 0.9394$ and $p = 0.9055$, $p = 0.5248$ and $p = 0.0850$, $p = 0.0013$ and $p = 0.0003$, $p = 0.2907$ and $p = 0.0026$, respectively), an unpaired two-tailed $t$-test was used. *$p < 0.05$, **$p < 0.01$, ***<0.001, NS = not significant. **d** Polymerase activities were determined in mock, APL-16-5 or APL-16-2 treated A549-5Ps cells carrying the viral mini-genome replicon ($n = 3$). Ribavirin was used as a positive control. Bar = mean. Error bars = ±SEM. For DMSO vs. Ribavirin, APL-16-2-2, 10 µM, APL-16-5-2, 10 µM): ($p = 0.0018$, $p = 0.0701$, $p = 0.0598$, $p = 0.0010$ and $p = 0.0002$, respectively), an unpaired two-tailed $t$-test was used. **$p < 0.01$, ***$p < 0.001$, NS = not significant. **e** qRT-PCR analysis of viral RNA (mRNA, vRNA, cRNA) from A/WSN/33-infected A549 cells following treatment with various doses of APL-16-5. Bar = mean. Error bars = ±SEM. For NP mRNA, cRNA, vRNA (DMSO vs. Ribavirin, APL-16-5-2, 10 µM): ($p = 0.0013$, $p = 0.0011$ and $p = 0.0001$, $p = 0.0015$, $p = 0.0001$ and $p = 0.0001$, $p = 0.0018$, $p = 0.0004$ and $p = 0.0001$, respectively), an unpaired two-tailed $t$-test was used. **$p < 0.01$, ***$p < 0.001$. Source data are provided as a Source Data file.

expression of individual subunits, suggesting that the formation of an RdRp complex may have stabilized all components. Therefore, it is possible that by reducing the level of PA, APL-16-5 may have also decreased the levels of other RdRp components, as shown in Fig. 2a.

Since APL-16-5 did not affect the expression of PA mRNA (Supplementary Fig. 2c), we suspected that APL-16-5 may have affected the stability of the PA protein. Indeed, western blot analysis revealed that APL-16-5 treatment led to a marked reduction in PA following the treatment of cells with the translation inhibitor cycloheximide (CHX) (Fig. 2c), whereas the level of PB1 was unchanged (Supplementary Fig. 2d). To quantify the decay kinetics of PA, we constructed a vector expressing PA-luciferase fusion protein, whose expression was sensitive to

APL-16-5 treatment and luciferase activity quantitatively reported the level of PA protein (Supplementary Fig. 2e). Using this reporter, we observed that APL-16-5 and another active compound APL-16-1, but not APL-16-2, reduced the half-life of PA protein from 18.8 h in control cells to 4.25 and 6.58 h in treated cells, respectively (Fig. 2d), suggesting that APL-16-5 and APL-16-1 accelerate the degradation of PA. Furthermore, the APL-16-5-induced destabilization of PA was reversed by the proteasome inhibitor MG132 (Fig. 2e), suggesting an important role of the proteasome in the depletion of PA protein by APL-16-5. This mechanism of action is supported by the results of the ubiquitination assay that showed that APL-16-5 induced the dose-dependent ubiquitination of PA (Fig. 2f), but not PB1 (Supplementary Fig. 2f). Thus, we concluded that APL-16-5

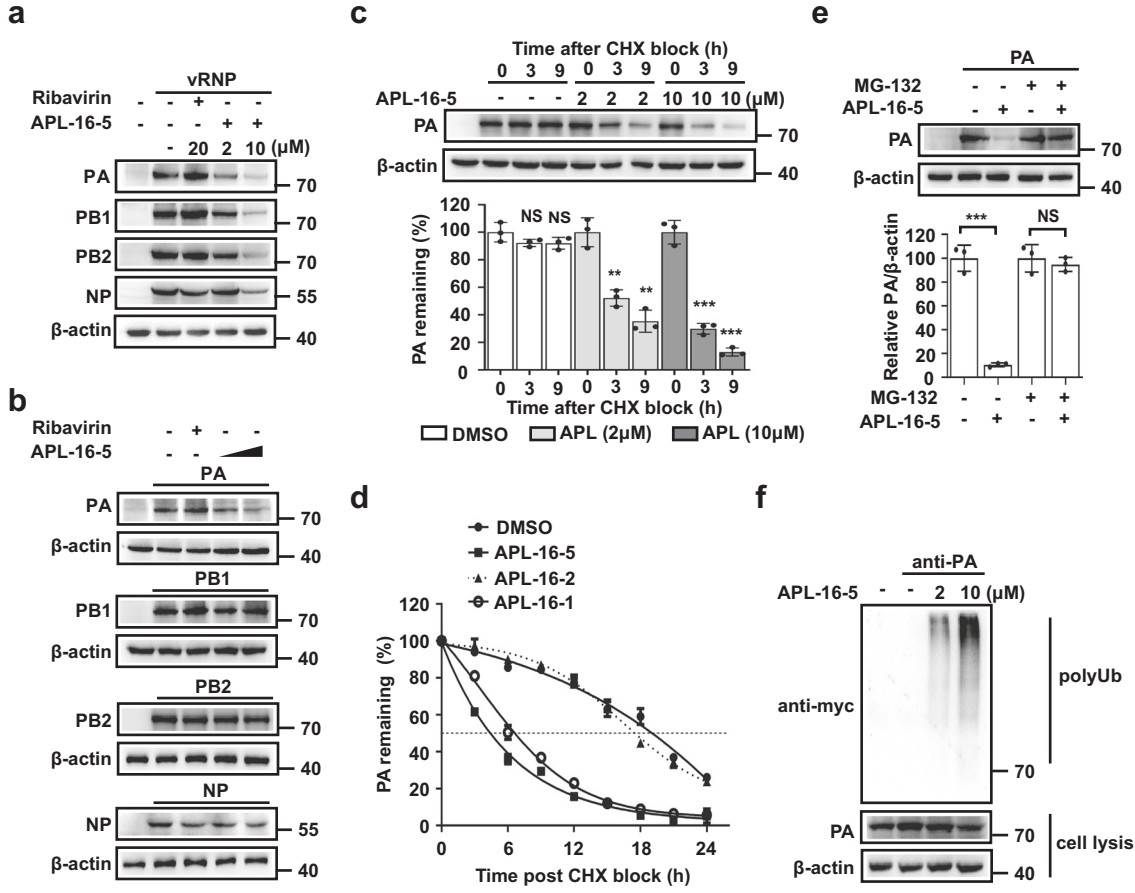

**Fig. 2 APL-16-5 induces proteasome-dependent degradation of PA. a** Western blot analysis of lysates from A549-5Ps cells following treatment with DMSO or APL-16-5. Ribavirin was used as a positive control. **b** Western blot analysis of lysates from HEK293T cells following treatment with DMSO or APL-16-5 (2 μM or 10 μM) after transfection of plasmid *PA, PB1, PB2,* or *NP* individually. **c** Western blot analysis to determine PA protein expression in HEK293T cells transfected with *PA* and treated with various amounts APL-16-5 after CHX treatment for the indicated time intervals. Protein band intensity was determined using the NIH Image J program. Bar = mean. Error bars = ±SEM. For DMSO, APL-16-5-2μM, APL-16-5-10 μM (0 vs. 3 and 9 h): ($p = 0.1518$ and $p = 0.1679$, $p = 0.0024$ and $p = 0.0011$, $p = 0.0002$ and $p = 0.0001$, respectively), an unpaired two-tailed t-test was used. NS = not significant. **d** Luciferase activity was measured to determine the half-life of PA in cells transfected with the *PA-luc* reporter DNA in the presence of APL-16-5 (10 μM), APL-16-1 (10 μM) and APL-16-2 (10 μM) for the indicated time intervals ($n = 3$). Bar =mean. Error bars = ±SEM. **e** Western blot analysis of HEK293T cell lysates following transfection with *PA* and treatment with APL-16-5 (10 μM), followed by DMSO or MG-132 (5 μM) for 6 h. Protein band intensity was determined using the Image J program. Bar = mean. Error bars = ±SEM. For MG-132(−, +) (DMSO vs. APL-16-5): ($p = 0.0002$ and $p = 0.5271$, respectively), an unpaired two-tailed t-test was used. ***$p < 0.001$, NS = not significant. **f** Immunoprecipitation and immunoblot analysis of HEK293T cells transfected with plasmid *PA* and *CW7-myc-ubiquitin* in the presence of increasing concentrations of APL-16-5. Cell lysates were subjected to IP using the anti-PA antibody and immunoblotted with anti-Myc antibody. Source data are provided as a Source Data file.

specifically induces the proteasome-dependent degradation of PA by inducing PA ubiquitination.

**IAV develops resistance to APL-16-5 by mutating PA.** To explore the possibility that the viral PA protein is the target of APL-16-5, we examined the effect of PA overexpression on the anti-IAV activity of APL-16-5. The results showed that increasing the expression of the PA protein, but not the PB1 protein, gradually alleviated the inhibitory effect of APL-16-5 on IAV replication (Fig. 3a and Supplementary Fig. 3a). This rescue effect by PA overexpression was also observed for APL-16-1 (Supplementary Fig. 3b), suggesting the PA-specific anti-IAV activity for both APL-16-5 and APL-16-1, because no such effect was observed for the antiviral effects of ribavirin (Supplementary Fig. 3c). These data suggest that reducing expression of the viral PA protein by APL-16-5 and APL-16-1 contributes significantly to their anti-IAV activity.

To further demonstrate that APL-16-5 targets PA, we selected APL-16-5-resistant IAV by passaging IAV in the presence of

APL-16-5. The selected IAV showed a 7.5-fold increase in the $EC_{50}$ of APL-16-5 compared with that of the wild-type IAV (Supplementary Fig. 3d), but remained fully susceptible to ribavirin (Fig. 3b). We then sequenced the APL-16-5-resistant IAV and identified two common mutations, N228K and AS704-705H (a mutant containing A704H followed by the replacement of S705 with a stop codon, resulting in the loss of six amino acids from the C-terminus of PA) of the PA protein (Supplementary Fig. 3e). Next, we introduced two mutations, N228K and AS704-705H (briefly A704H), either individually or in combination, into the wild-type PA gene and showed that the IAV with both N228K and A704H was resistant to APL-16-5 inhibition (Fig. 3c). This PA double mutant was completely resistant to APL-16-5-induced degradation (Fig. 3d). The single mutant N228K or A704H expressed at lower levels, and showed moderate resistance to degradation by APL-16-5. Collectively, these results suggest that the viral PA protein is at least one of the primary targets of APL-16-5. We noted that the drug-resistant viral mutant was still inhibited by high concentration of APL-16-5 albeit to a much less extent (Fig. 3b, c), although the

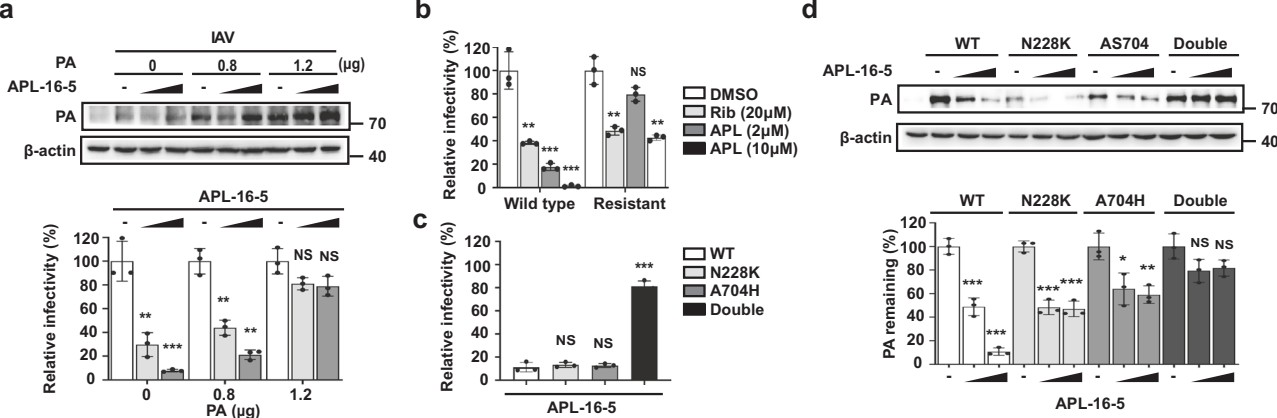

**Fig. 3 APL-16-5 inhibits Influenza A virus (IAV) by targeting PA for degradation. a** HEK293T-Gluc cells were transfected with increasing concentrations of *PA* plasmid DNA, followed by A/WSN/33 infection at a multiplicity of infection (MOI) of 0.5 and treatment with APL-16-5 (2 or 10 μM) for 24 h, then Gluc activity was determined and PA levels were assessed by western blot. For PA (0 μg vs. 0.8 and 1.2 μg): ($p = 0.0034$ and $p = 0.0007$, $p = 0.0015$ and $p = 0.0015$, $p = 0.0505$ and $p = 0.0541$, respectively). **b** HEK293T-Gluc cells were infected with wild-type or mutant virus at a multiplicity of infection (MOI) of 0.5 and treated with APL-16-5 or ribavirin at the indicated concentrations for 24 h, then Gluc activity was determined. Bar = mean. Error bars = ±SEM. For WT (DMSO vs. Ribavirin, APL-16-5-2, 10 μM): ($p = 0.0026$, $p = 0.0009$ and $p = 0.0004$, respectively), resistant (DMSO vs. Ribavirin, APL-16-5-2, 10 μM): ($p = 0.0020$, $p = 0.0564$ and $p = 0.0012$, respectively). **c** Antiviral activity of APL-16-5 was determined by measuring Gluc activity in HEK293T-Gluc cells that were infected with wild-type or mutated PA WSN virus at an MOI of 0.5 in the presence of APL-16-5 (10 μM). For APL-16-5 (WT vs. 228K, A704H, Double): ($p = 0.5032$, $p = 0.6055$ and $p = 0.0001$, respectively). **d** Western blot analysis of lysates from HEK293T cells that were transfected with wild-type or mutated *PA* DNA in the presence of increasing concentrations of APL-16-5 (2 or 10 μM). For WT, N228K, A704H, Double (DMSO vs. APL-16-5-2, 10 μM): ($p = 0.0009$ and $p = 0.0001$, $p = 0.0003$ and $p = 0.0003$, $p = 0.0241$ and $p = 0.0065$, $p = 0.0687$ and $p = 0.0666$, respectively). (**a-d**) The graph summarizes $n = 3$ independent experiments. Error bars show mean ± SEM (unpaired two-tailed *t*-test). **$p < 0.01$, ***$p < 0.001$, NS = not significant. Source data are provided as a Source Data file.

mutated PA was almost fully resistant to degradation by APL-16-5 (Fig. 3d), which suggests that APL-16-5 may act on more targets than PA to achieve its potent antiviral activity.

**APL-16-5-induced degradation of PA is dependent on TRIM25.** To determine the cellular mechanism underlying APL-16-5-induced PA degradation, SPR-LC-MS/MS was used to screen cellular proteins that bind to APL-16-5, as described previously[15,16]. Briefly, APL-16-5 was immobilized on a sensor chip and incubated with lysates of A549 cells followed by SPR and mass spectrometry analysis to identify and rank host proteins that potentially interact with APL-16-5. Among 79 total hits (score >200), five candidates were enriched in the protein degradation pathway (Fig. 4a). Next, we knocked down each of these genes using specific small interfering RNAs (siRNAs) and found that the inhibition of IAV by APL-16-5 was markedly abolished in cells transfected with siRNA targeting TRIM25 but not the other genes (Fig. 4b and Supplementary Fig. 4a). This suggested that TRIM25 is required for the anti-IAV activity of APL-16-5. We also used CRISPR/Cas9 to generate a *TRIM25*-knockout cell line, and observed that the anti-IAV activity of APL-16-5, but not that of ribavirin, was dependent on TRIM25 (Fig. 4c). Supporting these observations, overexpression of TRIM25 in *TRIM25*-knockout cells restored APL-16-5 inhibition of IAV close to that in the parental cells (Fig. 4d). We also noted that overexpressing TRIM25 by transfection of *TRIM25* DNA increased the inhibition of IAV by APL-16-5 (Supplementary Fig. 4b). Together, these data demonstrate a key role of TRIM25 in the anti-IAV activity of APL-16-5.

TRIM25 is an E3 ubiquitin ligase, suggesting that APL-16-5-mediated PA degradation may involve TRIM25-dependent PA ubiquitination. Indeed, TRIM25 knockout markedly inhibited APL-16-5-induced PA degradation (Fig. 4e). As a control, knockdown of another E3 candidate, HERC5, did not affect APL-16-5-induced PA degradation, indicating that HECR5 is not involved in the anti-IAV activity of APL-16-5 (Supplementary Fig. 4c). As expected, PA

underwent ubiquitination in wild-type cells but not in TRIM25 knockout cells, even with high concentrations of APL-16-5 (Fig. 4f). These data suggest that the cellular E3 ligase TRIM25 is required for APL-16-5-induced ubiquitination and PA degradation, and thus plays an essential role in the anti-IAV activity of APL-16-5. This essential role of TRIM25 does not preclude the involvement of other cellular factors in APL-16-5-induced PA degradation, which awaits further investigation.

**APL-16-5 induces ubiquitination of PA by engaging TRIM25.** The substrate specificity of each E3 ligase is often determined by their interaction. Therefore, we investigated whether APL-16-5 induces the ubiquitination of PA by promoting the interaction between TRIM25 and PA. First, we performed co-immuno-precipitation and noted a low basal level association of TRIM25 with PA, this association increased with increasing concentrations of APL-16-5 (Fig. 5a). The low level of PA found in the immuno-precipitated materials may represent either a non-specific binding or an indirect association in other complex. To validate this observation in cells, we performed an in situ proximity ligation assay (PLA), in which protein interactions are detected as individual fluorescent dots. Again, interactions between TRIM25 and PA were only detected in cells treated with APL-16-5 (Fig. 5b, upper panel). When MG132 was used to block the degradation of PA by APL-16-5, more and brighter fluorescent dots were detected (Fig. 5b, lower panel), accompanied by increased levels of PA (Supplementary Fig. 5a). Quantification of fluorescence revealed a four-fold increase in the colocalization between TRIM25 and PA as a result of MG132 treatment (Fig. 5c). In contrast, a co-immunoprecipitation assay showed that APL-16-5 was unable to promote the interaction of TRIM25 with double PA mutant (Supplementary Fig. 5b), which was shown resistant to APL-16-5 inhibition (Fig. 3c) and APL-16-5-induced PA degradation (Fig. 3d). These data suggest that APL-16-5 function by promoting the interaction of PA with the E3 ligase TRIM25 and render PA a substrate for ubiquitination

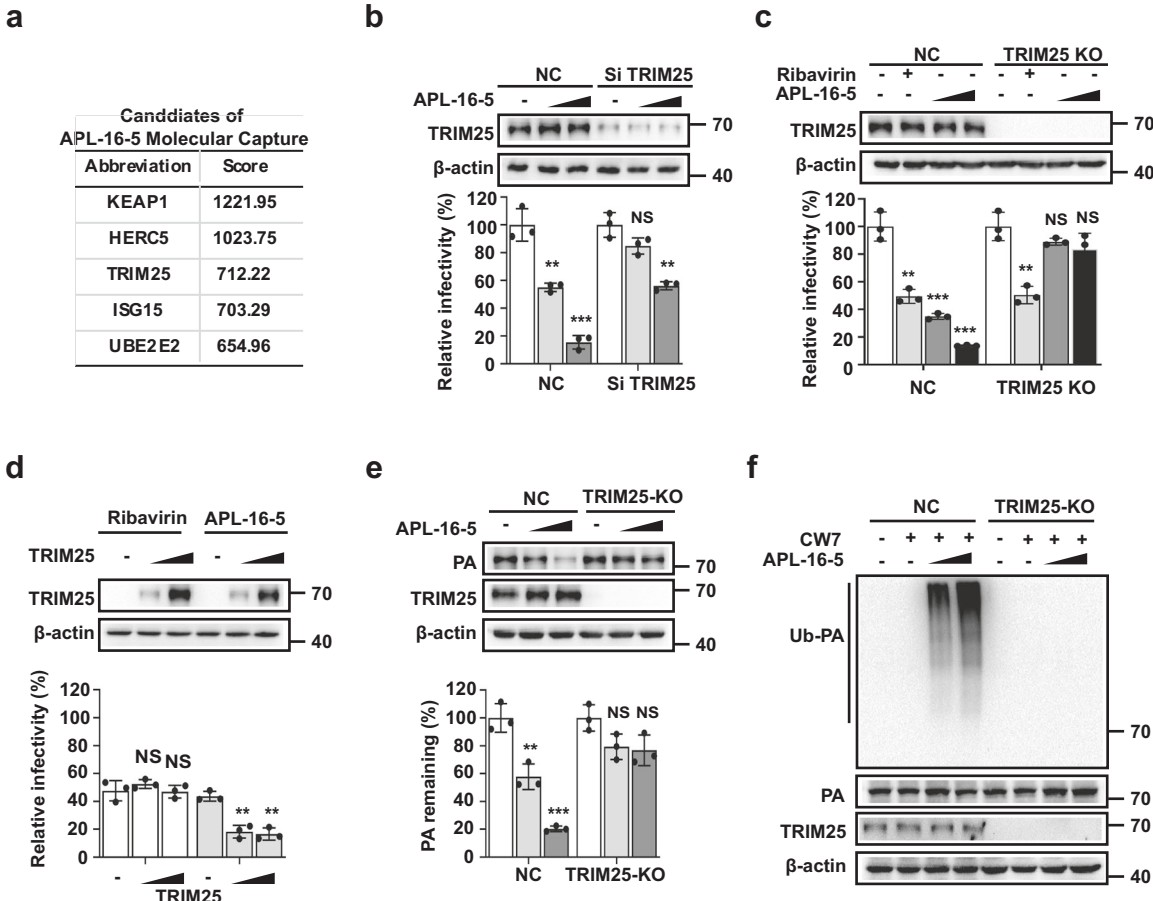

**Fig. 4 The anti-influenza A virus (IAV) activity of APL-16-5 is dependent on TRIM25. a** Schematic representation of the APL-16-5 molecular capture protocol and summary of ubiquitin-related candidates. **b** *TRIM25*-knockdown cells were infected with IAV A/WSN/33 in the presence of increasing amounts of APL-16-5 (2 μM or 10 μM). Virus production was determined via HEK293T-Gluc assay. TRIM25 levels were confirmed by western blot. For NC (DMSO vs. APL-16-5-2, 10 μM): ($p = 0.0031$ and $p = 0.0003$, respectively), TRIM25 KD (DMSO vs. APL-16-5-2, 10 μM): ($p = 0.0695$ and $p = 0.0013$, respectively). **c** *TRIM25*-knockout cells were transfected with increasing amounts of *TRIM25* and infected with A/WSN/33 in the presence of APL-16-5 (2 or 10 μM). Virus production was determined via HEK293T-Gluc assay. TRIM25 levels were determined by western blot. For NC (DMSO vs. Ribavirin, APL-16-5-2, 10 μM): ($p = 0.0017$, $p = 0.0005$ and $p = 0.0001$, respectively), TRIM25 KO (DMSO vs. Ribavirin, APL-16-5-2, 10 μM): ($p = 0.0020$, $p = 0.1457$ and $p = 0.1399$, respectively). **d** HEK293T cells were transfected with increasing amounts of *TRIM25* and infected with WSN/33 in the presence of APL-16-5 (2 or 10 μM). Virus production was determined via HEK293T-Gluc assay. For Ribavirin (TRIM25 0 ng vs. 200, 500 ng): ($p = 0.3106$ and $p = 0.9464$, respectively), APL-16-5 (TRIM25 0 ng vs. 200, 500 ng): ($p = 0.0117$ and $p = 0.0118$, respectively). **e** Western blot analysis of lysates from *TRIM25*-knockout cells transfected with *PA* DNA in the presence of increasing concentrations of APL-16-5 (2 or 10 μM). For NC (DMSO vs. APL-16-5-2, 10 μM): ($p = 0.0061$ and $p = 0.0002$, respectively), TRIM25 KO (DMSO vs. APL-16-5-2, 10 μM): ($p = 0.0528$ and $p = 0.0507$, respectively). **f** Immunoprecipitation and immunoblot analysis of *TRIM25*-knockout cells that were transfected with *PA* and *CW7-myc-ubiquitin* DNA in the presence of increasing concentrations of APL-16-5 (2 or 10 μM). **b–e** Data are normalized to those of the control group, which is arbitrarily set to 100%. The graph summarizes $n = 3$ independent experiments. Error bars show mean ± SEM (unpaired two-tailed *t*-test). **\*\*$p < 0.01$, \*\*\*$p < 0.001$, NS = not significant. Source data are provided as a Source Data file.

by TRIM25. Of note, results of PLA showed interactions of TRIM25 with PA (Supplementary Fig. 5c), but not with PB1 in cells expressing PA, PB1 and PB2, suggesting that PA but not the other subunits of viral polymerase complex is targeted by APL-16-5.

To further define the mechanism of APL-16-5 action, we examined the binding affinity of APL-16-5 to purified TRIM25 and PA using a bio-layer interferometry (BLI) kinetic binding assay. The results showed that APL-16-5 bound to both TRIM25 and PA in a concentration-dependent manner, with fitted $K_D$ values of 18 and 11 μM for TRIM25 and PA, respectively (Fig. 5d, e). Similarly, another anti-IAV compound, APL-16-1, bound to both TRIM25 and PA, whereas APL-16-2 bound to PA but not to TRIM25 (Supplementary Fig. 5d, e). Importantly, in the BLI assay, TRIM25 only bound to PA in the presence of APL-16-5, whereas APL-16-2 had no effect (Fig. 5f), further supporting the heterobifunctional nature of APL-16-5 to link TRIM25 and PA.

Moreover, the results of the in vitro ubiquitination assay showed that PA was polyubiquitinated by TRIM25 in the presence of APL-16-5, rather than APL-16-2 (Fig. 5g). This suggested that the APL-16-5-mediated interaction between TRIM25 and PA is sufficient to trigger the ubiquitination of PA by TRIM25. Interestingly, when cells were treated with both APL-16-5 and APL-16-2, PA levels were restored to those observed in the untreated cells (Fig. 5h). Since APL-16-2 also binds to PA, it must have diminished the binding of APL-16-5 to PA through competition, thus abrogating APL-16-5-mediated PA degradation. This finding further supports that APL-16-5 functions via a heterobifunctional mechanism that requires interaction with both the E3 ligase and substrate. It should be noted that APL-16-5 failed to promote the interaction of TRIM25 with the double PA mutant in a Co-IP assay (Supplementary Fig. 5b), which provides indirect evidence on the action of APL-16-5 and the resistant

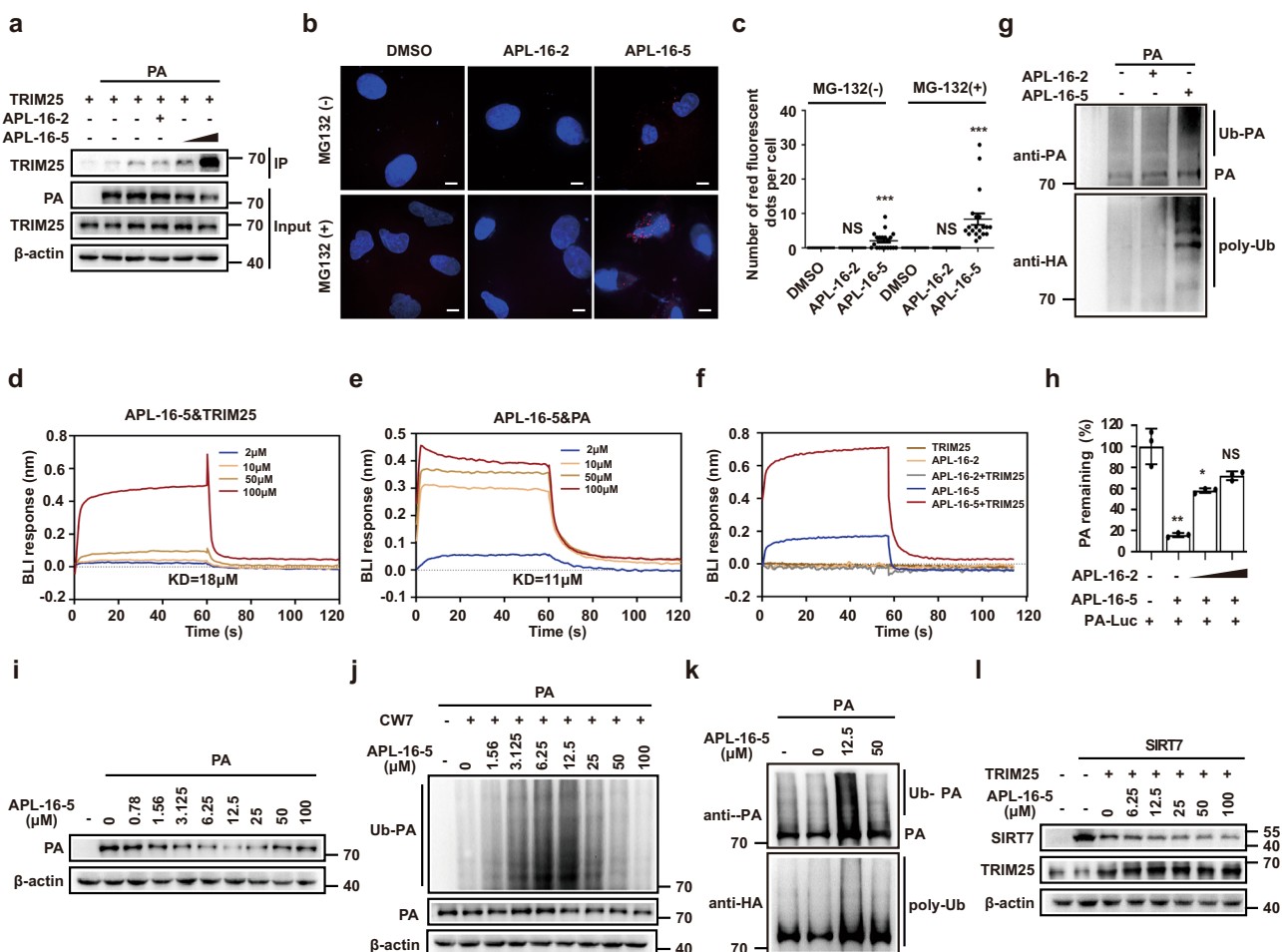

**Fig. 5 APL-16-5 links TRIM25 to PA and induces PA ubiquitination. a** Immunoprecipitation analysis of HEK293T cells transfected with the indicated concentrations of plasmid DNA, and treated with APL-16-5 (2 or 10 μM) or APL-16-2 (10 μM). **b** In situ PLA was performed to detect the interaction between TRIM25 and PA in the presence of APL-16-5 (10 μM) or APL-16-2 (10 μM), with or without MG-132 (5 μM). Representative confocal images are shown. The scale bars represent 5 μm. **c** Quantification of PLA red fluorescence signals from 20 cells. For MG-132(−, +) (DMSO vs. APL-16-2, APL-16-5): ($p = 1$ and $p = 0.0001$, $p = 1$ and $p = 0.0001$, respectively). **d** and **e** In vitro binding of APL-16-5 to purified TRIM25 (**d**) or PA (**e**) was determined by BLI binding assay. Representative association and dissociation sensorgrams are shown. **f** In vitro binding of purified TRIM25 to PA in the presence of APL-16-5 was determined by a BLI binding assay. Representative association and dissociation sensorgrams are shown. **g** In vitro ubiquitination of PA protein in the presence of purified TRIM25 and APL-16-5 (10 μM) or APL-16-2 (10 μM) was detected by western blot. **h** APL-16-2 competes with APL-16-5 for binding to PA. HEK293T cells were transfected with a PA-luc plasmid in the presence of APL-16-5 (10 μM) and APL-16-2 (50 μM and 100 μM). For APL-16-2 (DMSO vs. 0, 50, 100 μM): ($p = 0.0010$, $p = 0.0129$ and $p = 0.0050$, respectively). **i** Western blot of lysates from HEK293T cells following treatment with DMSO or APL-16-5 after transfection of plasmid PA. **j** Immunoprecipitation analysis of HEK293T cells transfected with plasmid PA and CW7-myc-ubiquitin in the presence of increasing concentrations of APL-16-5. **k** In vitro ubiquitination of PA protein in the presence of purified TRIM25 and an increasing concentration of APL-16-5. **l** Western blot of lysates from HEK293T cells following DMSO or APL-16-5 treatment after transfection of plasmid SIRT7 and TRIM25. **c** and **h** The graph summarizes $n = 3$ independent experiments. Error bars show mean ± SEM (unpaired two-tailed t-test). *$p < 0.05$, **$p < 0.01$, ***$p < 0.001$, NS = not significant. Source data are provided as a Source Data file.

mechanism. Measuring the affinity of PA mutants to APL-16-5 should provide more direct support.

To further test whether APL-16-5 acts as a bifunctional molecule, we performed a dose-response study of PA degradation and ubiquitination at high concentrations of APL-16-5, to determine if hook effect occurs. Hook effect is specific for bifunctional molecules which bind to both E3 ligase and substrate; thus at high ligand concentrations, individual binary complexes become saturated, which impedes the formation of the E3 ligase–substrate–ligand ternary complex and causes a loss of substrate degradation. In contrast, molecular glues have no measurable affinity for the free substrate, thus do not exhibit hook effect. The results showed that APL-16-5 gradually lost its activity of reducing the level of PA (Fig. 5i) and inducing PA ubiquitination in cells (Fig. 5j) and in cell-free assays (Fig. 5k) at high concentrations, which further supports

a biofunctional mechanism. In contrast, APL-16-5 did not affect the degradation of SIRT7 by TRIM25 (Fig. 5l), which most likely rules out the possible non-specific inhibition of TRIM25-mediated protein degradation at high concentrations of APL-16-5. Together, these data suggest that APL-16-5 acts by binding to both TRIM25 and PA, and induces TRIM25-mediated ubiquitination and subsequent degradation of PA.

**APL-16-5 protects mice from lethal IAV infection.** Finally, we investigated whether APL-16-5 could protect mice from lethal IAV infection. First, we treated rodent BHK21 cells with APL-16-5 and observed a substantial decrease in expression of the viral PA protein (Supplementary Fig. 6). This suggests that APL-16-5 can target IAV PA protein in mice as a result of the high homology between

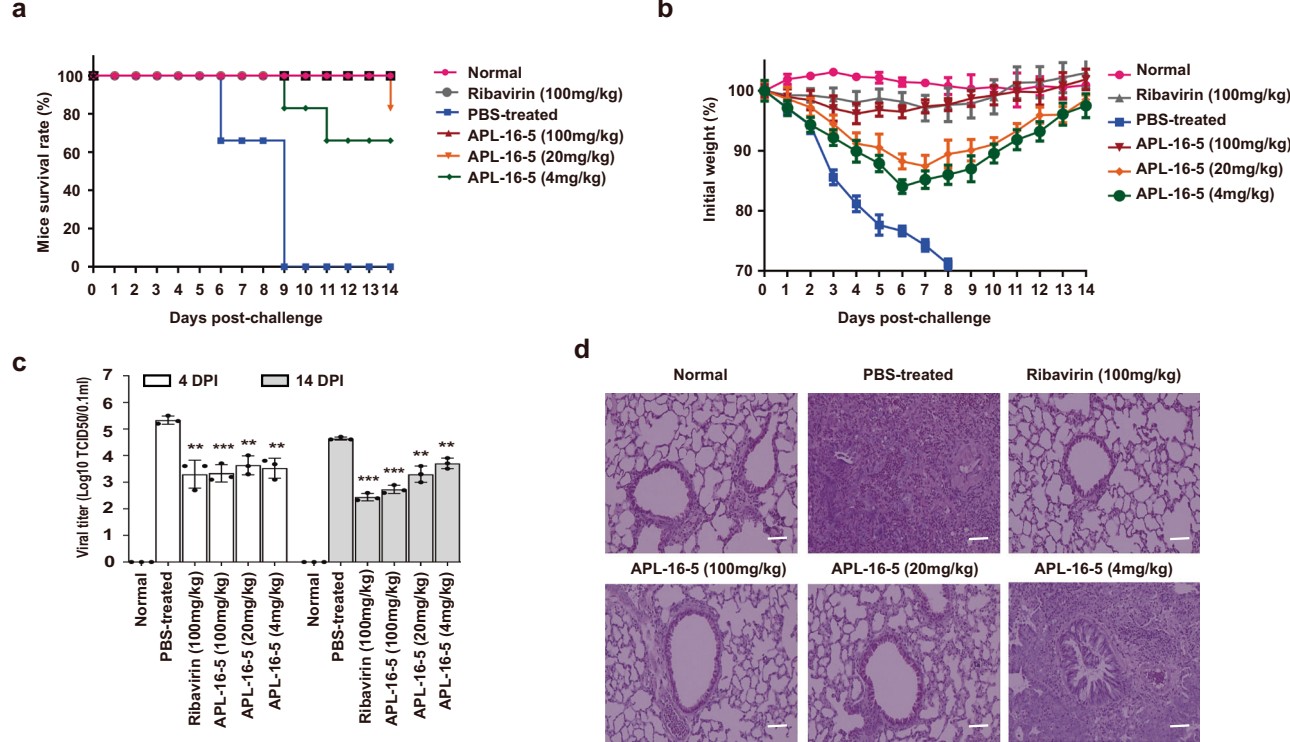

**Fig. 6 APL-16-5 protects mice from lethal influenza A virus (IAV) infection.** Mice (six per group) were infected with IAV WSN/33 ($5 \times LD_{50}$, in 50 µL PBS), and then treated with either APL-16-5 (4, 20, or 100 mg/kg), ribavirin (100 mg/kg), or PBS daily from day 1 to 8 post-infection. **a** Survival rate ($n = 6$), **b** body weight ($n = 6$). In a separate animal study of the same designs, **c** viral titers ($TCID_{50}$) in the lungs ($n = 3$) and **d** histopathology (one out of three shown, 200 X) were determined. Bar = mean. Error bars = ±SEM. For **c** viral titers at 4 and 14 DPI or until the death of mice in PBS-treated group (PBS-treated vs. Ribavirin, APL-16-5-100, 20, 4 mg/kg): ($p = 0.0029$, $p = 0.0006$, $p = 0.0015$ and $p = 0.0015$, $p = 0.0001$, $p = 0.0001$, $p = 0.0016$, and $p = 0.0015$, respectively), an unpaired two-tailed $t$-test was used. **$p < 0.01$, ***$p < 0.001$. Source data are provided as a Source Data file.

human and mouse TRIM25[17]. Next, we inoculated mice with a lethal dose of IAV followed by different doses of APL-16-5, administered orally on day 1 post-infection and then daily for 8 days. Remarkably, 100% of the mice treated with 100 mg/kg APL-16-5 and 67% of mice treated with a low dose of APL-16-5 (4 mg/kg) survived IAV infection, while none of the mice treated with PBS survived (Fig. 6a). APL-16-5 treatment increased the survival of mice, reaching 75% (Supplementary Table 2). Among the infected mice, no body weight loss was observed among those treated with 100 mg/kg of APL-16-5. The mice treated with lower doses of APL-16-5 (4 and 20 mg/kg) lost body weight, but to a much lesser degree compared with PBS-treated animals (Fig. 6b). Moreover, viral titers in the lungs were significantly reduced on days 4 and 14 post-infection in mice treated with APL-16-5, especially at doses of 20 and 100 mg/kg (Fig. 6c). Histopathological examination by hematoxylin and eosin (H&E) staining revealed that APL-16-5 treatment profoundly reduced the severity of pulmonary inflammation (Fig. 6d). Consequently, APL-16-5 significantly improved the lung index of IAV-infected mice, with 40–80% inhibition compared with that in the PBS-treated mice (Supplementary Table 3). Collectively, these results demonstrate strong protection of mice from lethal IAV infection by APL-16-5.

## Discussion

The results of this study demonstrate that APL-16-5 exerts potent anti-IAV activity in both cultured cells and mice. Specifically, APL-16-5 binds to both the E3 ligase TRIM25 and viral protein PA, and allows TRIM25 to ubiquitinate PA that then undergoes degradation in the proteasome. As a key subunit of the IAV RdRp complex, degradation of PA impairs viral RNA synthesis and consequently

inhibits IAV replication. We could not exclude the possibility that APL-16-5 may exert its antiviral activity by more than one mechanisms, especially in light of the much lower concentration of APL-16-5 required to inhibit IAV infection than degrading PA in the transient transfection experiment. This discrepancy can be attributed to the lower PA level in IAV-infected cells than that from the transfected plasmid DNA. Despite this difference, the important contribution of reduced PA expression by APL-16-5 to its anti-IAV activity is strongly supported by our observation increasing the expression of the PA protein alleviated the inhibitory effect of APL-16-5 on IAV replication (Fig. 3a).

To evaluate the selectivity of protein degradation by APL-16-5, we have characterized the protein profiles in cells treated with APL-16-5, using a multi-omics approach (Supplementary Tables 4–6). Only the level of FN1 protein was reduced by more than 2-fold in a TRIM25-dependent fashion at the post-transcriptional level. 12 more host proteins showed a moderate reduction ($-0.5 > \log2 > -1$) (Supplementary Table 7). These data demonstrate that APL-16-5-mediated protein down regulation is high selective, not promiscuous.

TRIM25 regulates the RIG-I-mediated IFN pathway[18,19], leading to antiviral response to influenza and other RNA viruses. APL-16-5 does not appear to interfere with the cellular function of TRIM25, since APL-16-5 did not affect the IFN response in cells (Supplementary Fig. 7). In addition, it was well tolerated by mice, with no signs of toxicity up to 500 mg/kg, suggesting the low toxicity of APL-16-5 and its potential use in patients. A direct antiviral role of TRIM25 was recently reported in a study showing that TRIM25 inhibits IAV RNA synthesis by directly binding to viral ribonucleoproteins and blocking viral RNA elongation, and this antiviral function of TRIM25 is independent of its E3 ligase activity[20]. Since the anti-IAV action of APL-16-5 requires the

ubiquitin ligase function of TRIM25, APL-16-5 should not act by enhancing the binding of TRIM25 to IAV ribonucleoproteins. We have thus engineered a strategy of using small molecules to endow TRIM25 with a new antiviral ability of degrading key viral enzymes by exploiting its E3 ligase function.

Three asperphenalenone compounds APL-16-5, APL-16-2, and APL-16-1 tested herein are composed of a linear diterpene derivative linked to a phenalenone derivative via a C–C bond, which shares the identical phenalenone structural fragment with different diterpene structural fragment (especially the terminal part of the diterpene). The results of BLI showed that all three compounds bound to PA with similar fitted KD values, whereas APL-16-5 and APL-16-1, but not APL-16-2, interacted with TRIM25. This suggests that the phenalenone fragment, shared by all three compounds, mainly contributes to their interaction with PA, and the diterpene fragment determines the binding to TRIM25 (different structures with different binding ability), while the detailed mechanism awaits further investigation.

The action of APL-16-5 conforms to the mechanism of a proteolysis-targeting chimera (PROTAC). PROTACs utilize the natural ubiquitin-proteasome system to chemically induce targeted protein degradation[21,22]. PROTACs recognize a target protein and then recruit an E3 ubiquitin ligase to trigger its ubiquitination and subsequent degradation[23,24]. Examples include ARV-110 that targets the androgen receptor and ARV-471 that targets the estrogen receptor[19,25]. In vitro data support the feasibility of inhibiting hepatitis B and C viruses by PROTAC-mediated degradation of viral proteins[26,27]. Our data suggest that APL-16-5 may act as an antiviral PROTAC. More experiments such as solving the ternary structure of the TRIM25/APL-16-5/PA complex are expected to provide definitive evidence on this concept.

Substantial progress has been made in the chemistry of hetero-bifunctional PROTAC. However, the in vivo activity of hetero-bifunctional PROTAC has not been extensively tested, and very few agents including ARV-110 have entered clinical studies[28–30]. The evidence provided in this study suggest that a microbial metabolite could act as a natural PROTAC to inhibit IAV replication. Given the structural diversity and biological activities of microbial metabolites, our findings opens us the possibility of using such a high source of compounds for the discovery of new PROTACs.

## Methods

**Cell lines and viral strains**. HEK293T (ATCC, CRL-3216), A549 (ATCC, CRL-185), MDCK (ATCC, CRL-34), Huh7.5.1 cells (Dr. Rongtuan Lin, McGill University), Vero cells (ATCC, CCL-81) and BHK21 (ATCC, CCL-10) cells were maintained in Dulbecco's modified Eagle's medium (DMEM; Gibco) supplemented with 10% (v/v) fetal bovine serum (FBS; Gibco), at 37 °C in a 5% $CO_2$ incubator. HEK293T-Gluc and A549-5Ps cells were generated as previously described[13]. HEK293T-Gluc constitutively expresses the negative-strand RNA of the *Gluc* gene that is converted into positive-strand RNA upon IAV infection and expresses Gluc. A549-5Ps cells contain an IAV mini-genome replicon by the transduction of A549 cells with lentiviral particles carrying *PB2, PB1, PA, NP*, and *Glu\*-/Bsd* genes. The *TRIM25*-knockout cell line was generated using CRISPR/Cas9. The gRNA target sequence was 5′-caccgTGGTAGACGGCGCGGCACTG-3′.

Influenza A/WSN/1933 (H1N1) was generated using the pHW2000 eight-plasmid system[13]. Single-round infectious IAV was produced with A/WSN/1933 by displacing the hemagglutinin (HA) coding sequence with the *Gluc* sequence. A/Puerto Rico/8/1934 (H1N1) was grown in embryonated chicken eggs. A/Beijing/30/95 (H3N2) was kindly provided by Dr. Wenjie Tan (China CDC). Influenza B virus strains B/Beijing-Haidian/1386/2013 and B/Massachusetts/02/2012 were kindly provided by Dr. Yuelong Shu (Chinese National Influenza Center). The infectious viral DNA clones JFH1 for producing infectious HCV were kindly provided by Takaji Wakita. ZIKV (strain FSS13025) was a kind gift from Dr. Mark Wainberg (McGill University). To generate mutant IAVs resistant to APL-16-5, influenza A/WSN/33 (H1N1) was passaged in MDCK cells with increasing concentrations of APL-16-5 (2, 2.5, 3, and 4 μM). After 18 passages, the resistant viruses were selected and genetically characterized by DNA sequencing of the viral PA gene.

**Compounds, plasmids, and antibodies**. The natural compounds tested in this study were isolated from *Aspergillus* spp. CPCC 400735 as previously described[12]. Ribavirin (#S2504), sofosbuvir (#S2794), and nucleozin (#S0433) were purchased

from Selleck Chemicals. The translation inhibitor CHX (#C8030) and proteasome inhibitor MG132 (#M8699) were purchased from Sigma-Aldrich and used to analyze PA stability and degradation.

The pHW181-PB2, pHW182-PB1, pHW183-PA, and pHW185-NP plasmids were kindly provided by Dr. Robert G. Webster (St. Jude Children's Research Hospital)[13]. *PA* mutants (N228K, AS704-705H, and N228K/AS704-705H) were generated in the context of a wild-type *PA* plasmid using the QuickMutation™ Site-Directed Mutagenesis Kit (Beyotime, #D0206) following the manufacturer's instructions. cDNA encoding Myc-tagged human ubiquitin (pRBG4-CW7-myc-ubiquitin) was kindly provided by Xiaofang Yu. Plasmids encoding FLAG-tagged TRIM25 were purchased from Youbio Biological Technology Co., Ltd. PA-luc was generated by cloning the nano-luciferase gene into the C-terminal of the wild-type PA protein. The pCMV6-SIRT7 plasmid was purchased from OriGene Technologies, Inc.

The following antibodies were used for western blotting or immunoprecipitation: rabbit anti-PA (1:1000, GTX125932), anti-NP (1:1000, GTX125989), anti-PB1 (1:1000, GTX125923), anti-PB2 (1:1000, GTX125925) from Genetex, mouse anti-β-actin (1:5000, ab8224), anti-core (1:5000, ab2740), and goat anti-Myc (1:1000, ab9132) from Abcam; goat anti-TRIM25 (1:3000, 610570) from BD Biosciences; goat anti-rabbit IgG horseradish peroxidase (HRP)-linked antibody (1:5000, ZB-2301), and anti-mouse IgG HRP-linked antibody (1:5000, ZB-2305) from Beijing Zhongshan Jinqiao Biotechnology; Alexa Fluor-conjugated secondary antibodies (1:2000, S11223) from Thermo Scientific. Anti-HA (1:1000, SC-7392) antibody was purchased from Santa Cruz Biotechnology. Anti-SIRT7 (1:1000, TA326876) antibody was from OriGene Technologies, Inc.

**Viral infection and TCID$_{50}$ assay**. Cells were incubated with influenza virus at a multiplicity of infection (MOI) of 0.5, for 1 h at room temperature and cultured for 24 or 48 h at 37 °C in fresh DMEM. Viral titers in supernatants were determined using MDCK cells by the 50% tissue culture infectious dose (TCID$_{50}$) assay, using the method described by Reed and Muench[13].

**Gluc activity assay**. Briefly, before assay, the coelenterazine-h (Promega, #S2011) was diluted to 16.7 μM with phosphate-buffered saline (PBS). Cell culture supernatants were added to white and opaque 96-well plates, followed by the automated injection of 60 μL coelenterazine-h per well, in. Photon counts were acquired for 0.5 s, using a Centro XS3 LB 960 microplate luminometer (Berthold Technologies, BadWildbad, Germany). The data were collected using MikroWin 2010 software (version 5.17).

**Cytotoxicity analysis**. HEK293T, A549, and MDCK cells were cultured at 37 °C in 96-well plates for 24 h and then incubated with the test compounds at the indicated concentrations. Mock-treated cells served as control. A cell viability assay was performed after 48 h using Cell Counting kit-8 (CCK-8, Beyotime, #C0038).

**RNA isolation and quantitative RT-PCR (RT-qPCR)**. Total RNA was extracted from the infected cells using TRIzol reagent (Invitrogen, #15596-018), and cDNA was synthesized using M-MLV Reverse Transcriptase (Promega, #M1705) with oligo(dT) or specific primers. The levels of viral RNA were determined by RT-qPCR using the SYBR premix Ex Taq II kit (Takara, #RR820A), according to the manufacturer's instructions. The data were collected by QuanStudio Design& Analysis software (version 1.5.1) and were analyzed by normalizing against internal standards using the $2^{-\Delta\Delta Ct}$ method. Primer sequences for vRNA, mRNA, and cRNA were as follows: *NP* vRNA, reverse transcription primer 5′-GGCCGTCA TGGTGGCGAATGAATGGACGGAGA ACAAGGATTGC-3′; and real-time PCR forward 5′-CTCAATATGAGTGCAGACCGTGCT-3′; reverse, 5′-GGCCGTCA TGGTGGCGAAT-3′. *NP* mRNA, reverse transcription primer 5′-CCAGATCG TTCGAGTCGTTTTTTTTTTTTTTTTTTCTTTAATTGTC-3′; and re-time PCR forward 5′-CGATCGTGCCCTCCTTTG-3′; reverse 5′-CCAGATCGTTCGAGTC GT-3′. *NP* cRNA, reverse transcription primer 5′-GCTAGCTTCAGCTAGGCAT CAGTAGAAACAAGGGTATTTTTCTTT-3′ and reverse 5′-CGATCGTGCC CTCCTTTG-3′; reverse 5′-GCTAGCTTCAGCTAGGCATC-3′. *GAPDH* real-time PCR forward 5′-GTCCACTGGCGTCTTCACCA-3′, reverse 5′-GTGGCAGTG ATGGCATGGAC-3′. PA real-time PCR forward 5′-TGGGATTCCTTTCGTCA GTC-3′, reverse 5′-TGAGAAAGCTTGCCCTCAAT-3′.

**Immunofluorescence and confocal microscopy**. Cells were washed twice with ice-cold PBS and fixed in 4% paraformaldehyde before being permeabilized with 0.25% Triton X-100 for 10 min. Cells were blocked with 1% bovine serum albumin (BSA) for 1 h prior to incubation with primary antibodies overnight at 4 °C. After washing three times in 1 × PBS, cells were incubated with Alexa Fluor-conjugated secondary antibodies for 1 h with gentle shaking. Nuclei were stained with DAPI (4′,6′-diamidino-2-phenylindole dihydrochloride). Fluorescence was visualized using a PerkinElmer Ultra View VoX confocal imaging system.

**In situ PLA**. A Duolink® PLA kit (Sigma Aldrich, #DUO92101,) was used for the in situ PLA assay. Briefly, samples fixed in 4% paraformaldehyde were incubated with blocking solution to saturate nonspecific binding and then with primary antibodies at 37 °C for 1 h. Thereafter, the slides were incubated for 1 h at 37 °C with the Duolink® PLA probes. The ligation solution was then added and incubated

for 30 min at 37 °C. The ligation solution was removed with wash buffer A. The amplification solution was then added and samples were incubated for 100 min at 37 °C before being removed with wash buffer B. Finally, Duolink in situ mounting medium with DAPI was added to samples. Fluorescence was visualized using a PerkinElmer Ultra View VoX confocal imaging system.

**Immunoprecipitation (IP) and immunoblotting**. For IP, HEK293T cells were transfected using Lipofectamine 2000 (Thermo Scientific, #11668027) and treated with APL-16-5. After 48 h, cells were washed with PBS and lysed in buffer containing 25 mM Tris (pH 7.4), 150 mM NaCl, 1% NP-40, 1 mM EDTA, and 5% glycerol (Thermo Scientific, #87787) supplemented with a protease inhibitor cocktail (Roche). After 1 h on ice, the lysates were centrifuged at $16,200 \times g$ for 10 min at 4 °C to remove cell debris. Cellular extracts were incubated with antibodies for 4 h and then with protein A/G agarose (Beyotime, #P2055) overnight at 4 °C. The sepharose samples were washed five times with cell-lysis buffer, boiled with SDS loading buffer for 10 min, and subjected to western blot analysis. After separation by SDS–PAGE, proteins were transferred onto PVDF membranes (Millipore) and immunoblotted with the indicated primary and HRP-conjugated secondary antibodies. The membranes were incubated with the chemiluminescent HRP substrate (Millipore) and protein signals were determined using a Gel Doc XR + molecular imager (Bio-Rad).

**Bio-layer interferometry (BLI) binding assay**. Binding affinity and kinetic profiles were measured using Octet RED (ForteBio). The purified PA or TRIM25 was biotinylated using EZ-Link NHS-LC-LC-Biotin (Thermo Scientific, #21343). The biotinylated PA or TRIM25 (50 μg/mL) was then captured using super streptavidin (SSA) biosensors (120 s, at 30 °C, 100 g). A duplicate set of sensors was incubated in buffer (0.002% Tween-20, PBS, pH = 7.4) without protein to control for background binding. Both the ligand and reference biosensors were quenched with 5 μg/mL biotin for 1 min. To determine the KD, the binding of a diluted series of compounds was detected for 60 s association (kon, 1/Ms) followed by 60 s dissociation (kdis, 1/s) in parallel to the ligand and reference biosensors. In addition, blank binding cycles using only buffer were used to correct the baseline shift during the analysis. After measurements were obtained, a double reference subtraction method was used to subtract the effects of baseline drift and nonspecific binding. KD was acquired by fitting into a 1:1 binding model by global fitting of multiple kinetic traces and analyzed using Data Analysis software (version 9.0).

**In vitro ubiquitination assay**. Recombinant human C-myc/DDK TRIM25 protein (#TP303757), purified from HEK293T cells, was purchased from Origene Technologies, Inc. and recombinant C-His-PA was purified from HEK293T cells in our laboratory, both of whose purity was above 80% determined by SDS–PAGE gel and coomassie blue staining. TRIM25 constructs were incubated at 37 °C with recombinant human ubiquitination activating enzyme (E1, 100 nM), recombinant human UbcH5a/UBE2D1 (E2, 1 μM), and recombinant HA-ubiquitin (40 μM) in 950 mM Tris (pH 7.5), 150 mM NaCl, 0.5 mM TCEP, 5 mM ATP, and 10 mM MgCl$_2$. The reactions were stopped by adding SDS–PAGE sample buffer and boiling for 20 min. Immunoblotting was performed with anti-HA (1:2000) and anti-PA (1:2000) antibodies.

**APL-16-5 molecular capture experiments**. To identify and quantify APL-16-5 target host proteins, surface plasmon resonance (SPR) and high-performance liquid chromatography (HPLC) mass spectrometry (MS) experiments were performed. APL-16-5 was formulated with 50% dimethyl sulfoxide (DMSO) at a concentration of 100 mM. Continuous APL-16-5 was printed on a chip surface by auto-spotting three times using a BioDot™-1520 array printer (CA, USA). The chip surface was then printed with a $50 \times 50$ matrix of 18.75 μL (1.875 μmol) APL-16-5 sample in total, with 2.5 nL a projected target of the solution.

Cell lysis dosage calibration: After lysing A549 cells, the protein concentration of the sample was calibrated using a bicinchoninic acid (BCA) protein assay kit (Thermo Scientific3), leading to a concentration of 411.77 μg/mL. The concentration of the samples was adjusted to 200 g/mL using 1× lysis buffer.

Chip performance calibration: Each chip was fabricated by Lumera Co. Ltd. (Kaiserslautern, Germany). The difference in chip binding quantity between batches was <0.5%. The optimal resonance angle was automatically adjusted using a bScreen LB 991 biochip analyzer (Berthold Technologies, Germany).

Target protein capture: In the SPR assay, APL-16-5 was immobilized on the surface of the chip, and A549 cell lysate was used as the liquid phase. The APL-A549 sample curve indicated target protein binding in the area spotted with APL-16-5. The background curve shows signals in the non-spotted area.

Time procedures: Over 0–260 s, the system was pre-washed and the surface of the chip was soaked in running buffer. At this time, the resonance intensity was ~0 resonance units (RUs). During 260–520 s, APL-16-5 on the chip surface starts to capture the protein target in the lysate. Over 520–820 s, the chip was washed to remove non-specific adhesive proteins from the surface. Target protein-specific binding to APL-16-5 was retained on the surface of the chip. The resonance intensity decreased and plateaued (~607.50 RU). Non-specific binding of the non-point sample area was gradually cleaned, the resonance intensity of the background value gradually returned to the baseline level (~42.33 RU), and the chip background noise returned to normal levels.

**IAV infection of mice**. Four- to six-week-old female BALB/c mice were purchased from the Academy of Military Medical Sciences Laboratory (China) and housed in a room temperature (20 ± 2 °C) and humidity (50 ± 10%) mouse facility with free access to food and water under a 12 h dark/light cycle. All animal experiments were approved by the Institutional Animal Care and Use Committee of the Institute of Medicinal Biotechnology of the Chinese Academy of Medical Sciences. Mice (six per group) were intranasally infected with influenza A/WSN/33 virus ($5 \times LD_{50}$, in 50 μL PBS) or mock-infected, and then received either APL-16-5 (4, 20, or 100 mg/kg), ribavirin (100 mg/kg), or PBS daily on days 1–8 post-infection. Compound diluent was 0.5% Sodium Carboxymethylcellulose. Mice were monitored daily, and survival and weight loss were recorded up to day 14. In a separate animal study of the same designs, virus titers in the lungs were determined from mice (three mice from each group) sacrificed on day 4 and 14 post virus exposure or until animal died. Mouse lung was collected and homogenized. Each sample was assayed in triplicate for viral titers in MDCK cells with the TCID$_{50}$ method. Samples for histopathology examination were collected on day 14 after virus inoculation or until animal died. The lungs were fixed in 4% paraformaldehyde. Fixed sections of paraffin-embedded lungs were stained with H&E and imaged under a microscope (×200).

**ZIKV infectivity assay**. Titer of ZIKV stock was determined by infecting Vero cells, the results were calculated as the median tissue culture infective dose (TCID$_{50}$). Vero cells were infected with ZIKV at an MOI of 0.01. ZIKV infection was determined by quantifying viral RNA with qRT-PCR. Total cellular RNA was extracted 48 h post-infection with TRIzol reagent (Invitrogen, USA). cDNA was synthesized using Primescript RT Master Kit (Takara, Japan). Level of viral RNA was determined by qRT-PCR using SYBR premix Ex Taq II kit (Takara). ZIKV gene NS2A was amplified with primers 5′-CCACGCACTGATAACAT-3′ (forward) and 5′-AAGTAGCAAGGCCTGCTCT-3′ (reverse). Cellular GAPDH RNA was amplified with primer pair 5′-atca tccctgcctctactgg-3′/ 5′-gtcaggtccaccactgacac-3, the results served as internal controls to normalize ZIKV RNA data. For quantification, the 2−ΔΔCt method was used to calculate the relative ZIKV RNA levels against GAPDH.

**Quantitative proteomic analysis by LC–MS/MS**. The tryptic peptides were dissolved in solvent A (0.1% formic acid, 2% acetonitrile/in water), directly loaded onto a home-made reversed-phase analytical column (25-cm length, 75/100 μm i.d.). Peptides were separated with a gradient from 6% to 24% solvent B (0.1% formic acid in acetonitrile) over 70 min, 24–35% in 14 min and climbing to 80% in 3 min then holding at 80% for the last 3 min, all at a constant flow rate of 450 nL/min on a nanoElute UHPLC system (Bruker Daltonics).

The peptides were subjected to Capillary source followed by the timsTOF Pro (Bruker Daltonics) mass spectrometry. The electrospray voltage applied was 1.60 kV. Precursors and fragments were analyzed at the TOF detector, with a MS/MS scan range from 100 to 1700 $m/z$. The timsTOF Pro was operated in parallel accumulation serial fragmentation (PASEF) mode. Precursors with charge states 0–5 were selected for fragmentation, and 10 PASEF-MS/MS scans were acquired per cycle. The dynamic exclusion was set to 30.

The resulting MS/MS data were processed using MaxQuant search engine (version1.6.15.0). Tandem mass spectra were searched against human SwissProt database (Home-sapiens-9606-SP-20201214.fasta, 20395 entries) concatenated with reverse decoy database. Trypsin/P was specified as cleavage enzyme allowing up to two missing cleavages. The mass tolerance for precursor ions was set as 40 ppm in First search and 40 ppm in Main search, and the mass tolerance for fragment ions was set as 0.04 Da. Carbamidomethyl on Cys was specified as fixed modification, and acetylation on protein N-terminal and oxidation on Met, acetylation/succinylation/ubiquitylation/crotonylation/2-ohi-butyrrylation/malonylation/… on Lys/phosphorylation on Ser, Thr, Tyr were specified as variable modifications. TMT-6plex quantification was performed. FDR was adjusted to <1%. The cut-off values for identification of potential hits was set at log$_2$FoldChange < 0.5 and $p$ value < 0.05.

**RNA sequence**. Total RNA was extracted using the TRIzol reagent according to the manufacturer's protocol. RNA purity and quantification were evaluated using the NanoDrop 2000 spectrophotometer (Thermo Scientific, USA). RNA integrity was assessed using the Agilent 2100 Bioanalyzer (Agilent Technologies, Santa Clara, CA, USA). Then the libraries were constructed using TruSeq Stranded mRNA LT Sample Prep Kit (Illumina, San Diego, CA, USA) according to the manufacturer's instructions. The transcriptome sequencing and analysis were conducted by OE Biotech Co., Ltd. (Shanghai, China).

The libraries were sequenced on an Illumina HiSeq X Ten platform and 150 bp paired-end reads were generated. About 5.64–6.49 G raw reads for each sample was generated. Raw data (raw reads) of fastq format were firstly processed using Trimmomatic and the low-quality reads were removed to obtain the clean reads. Then about 37.15 G clean reads for each sample were retained for subsequent analyses. The clean reads were mapped to the human genome (GRCh38) using HISAT2. FPKM of each gene was calculated using Cufflinks, and the read counts of each gene were obtained by HTSeq-count. Differential expression analysis was performed using the DESeq (2012) R package. $p$ value < 0.05 and fold change < 0.5 was set as the threshold for significantly differential expression.

**Statistical analysis**. Data are presented as mean ± standard deviation (SD) from at least three independent experiments and evaluated using a two-tailed, unpaired Student's *t*-test via Prism (version 8.0, GraphPad Software, San Diego, CA, USA). Differences between the two indicated settings were considered statistically significant at $p < 0.05$(*), $p < 0.01$(**), and $p < 0.001$(***). NS denotes non-significant.

**Reporting summary**. Further information on research design is available in the Nature Research Reporting Summary linked to this article.

## Data availability

The human genome (GRCh38) data used in this study are available in the NCBI Genome database under accession code PRJNA31257. Sequence references used for tandem mass spectra in this study are available in human SwissProt database [https://www.uniprot.org/]. The names of the RNA-seq repository/respositories and accession number(s) can be found below: NCBI SRA; BioProject ID PRJNA811744. The mass spectrometry proteomics data have been deposited to the ProteomeXchange Consortium via the PRIDE[31] partner repository with the dataset identifier PXD031993. All relevant data supporting the findings of this study are available within the paper and/or Source data files and Supplementary Information File. Source data are provided with this paper.

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

## Acknowledgements

We thank the National Microbial Resource Center (No. NMRC-2020-3) and the CAMS Collection Center of Pathogenic Microorganisms (CAMS-CCPM-A) for providing valuable reagents. This work was supported by National Key Research and Development program of China 2016YFD0500307 (to S.C.), National Natural Science Foundation of China 81973220 (to L.Y.), 81971950 (to J.W.) and 82104250 (to J.Z.), Fundamental Research Funds for the Central Universities 33320200046 (to J.Z.), and CAMS Innovation Fund for Medical Sciences 2021-I2M-1-038 (to S.C.) and 2021-I2M-1-055 (to L.Y.).

## Author contributions

S.C. and L.Y. designed the study; J.Z. and J.W. performed the main part of the experiment; X.P., X.F., Z.G. W.L., and T.Z. performed the fermentation and prepared the compounds; Z.L., D.Y. Y.Z., and R.Z. performed the antiviral testing; Q.L. supervised the in vitro binding experiment; J.W., X.L., T.D., and F.G. contributed to the data analysis; J.W. prepared the manuscript; C.L. and S.C. revised the manuscript; S.C., L.Y., and F.G. conceived and supervised the project; all authors approved the final version of the manuscript.

## Competing interests

S.C., L.Y., J.Z., Y.Z., T.Z., X.P. and X.F. are inventors of the related patent/application 201710077812.X., which was filed by the Institute of Medicinal Biotechnology. This patent includes some in vivo antiviral data described herein, which does not impose any restrictions for the scientific use of the related compounds. The remaining authors declare no competing interests.
