## [Peer Review File · Nature Communications]

Referee #2**1. the authors should perform quantitative proteomics experiments to look at the selectivity of degradation by APL-16-5**

Answer: Thank you for this suggestion. We have investigated the protein profile in the cells treated with APL-16-5 using a multi-omics approach. Briefly, an LC-MS/MS-based label-free quantitative proteomics approach was used to analyze protein expression profile in control and TRIM25-knockout cells with or without APL-16-5 treatment. In total, 6503 proteins were detected by proteomics analysis (a full list of identified proteins is available upon request). Only one cellular protein (FN1) were significantly downregulated by more than $\log_2(-1)$ fold with a p value < 0.05 by APL-16-5 in a TRIM25-dependent manner. We also performed transcriptomic study using an Illumina HiSeq X Ten platform to identify the genome-wide transcriptional changes associated with APL-16-5 treatment. These data together showed that only FN1 was reduced by more than 2-fold, and 12 more host proteins showed a moderate reduction ($-0.5 > \log_2 > -1$) in a TRIM25-dependent fashion at the post-transcriptional level. Therefore, the expression of very few host proteins was affected by APL-16-5, which demonstrates the very high selectivity of APL-16-5-mediated degradation. We have now presented these data in the Extended Data Table S4-7, and described the results on page 17 as the following:

“To further assess the selectivity of degradation by APL-16-5, we have investigated the protein profile in the cells treated with APL-16-5 using a multi-omics approach (Extended Data Table S4-6), and found that only the level of FN1 was reduced by more than 2-fold in a TRIM25-dependent fashion at the post-transcriptional level, and 12 more host proteins showed a moderate reduction ($-0.5 > \log_2 > -1$) (Extended Data Table S7), demonstrating a high selectivity of APL-16-5 action.”

2. the authors claim that this compound is a PROTAC, but this compound could also be a molecular glue. Do the authors know that this is acting in a bifunctional manner versus a molecular glue-based mechanism? Do the authors know what part of the molecule is interacting with PA vs TRIM25?

Answer: Our data support that APL-16-5 acts as a bifunctional molecule. First, APL-16-5 binds to both the E3 ligase TRIM25 and the substrate PA (Fig. 5 and Extended Data Fig. S5), as opposed to a molecular glue that acts only by binding to the E3 ligase. Second, the compound APL-16-2, which binds to PA but not TRIM25, blocks PA degradation by APL-16-5 through competitive binding to PA (Fig. 5H), which supports the importance of binding to PA for APL-16-5-mediated degradation. Third, APL-16-5 has two ligand-binding moieties connected by a linker, which resembles a typical structure of heterobifunctional molecule (Fig. 1). Lastly, we have now performed a dose-response study of PA degradation and ubiquitination at high concentrations of

APL-16-5, to determine if hook effect occurs. Hook effect is specific for bifunctional molecules which bind to both E3 ligase and substrate; thus at high ligand concentrations, individual binary complexes become saturated, which impedes the formation of the E3 ligase-substrate-ligand ternary complex and causes a loss of substrate degradation. In contrast, molecular glues have no measurable affinity for the free substrate, thus do not exhibit hook effect. We observed that increasing the concentration of APL-16-5 gradually inhibited its activity of reducing the level of PA (Fig. 5i) and inducing PA ubiquitination in cells (Fig. 5j) and in cell-free assays (Fig. 5k). In contrast, APL-16-5 did not affect the degradation of SIRT7 by TRIM25 (Fig. 5l), which rules out non-specific inhibition of TRIM25-mediated protein degradation at high concentrations APL-16-5. We have now presented these data in the new Fig. 5, and described the results on page 14 as the following:

“To further test whether APL-16-5 acts as a bifunctional molecule, we performed a dose-response study of PA degradation and ubiquitination at high concentrations of APL-16-5, to determine if hook effect occurs. Hook effect is specific for bifunctional molecules which bind to both E3 ligase and substrate; thus at high ligand concentrations, individual binary complexes become saturated, which impedes the formation of the E3 ligase–substrate–ligand ternary complex and causes a loss of substrate degradation. In contrast, molecular glues have no measurable affinity for the free substrate, thus do not exhibit hook effect. The results showed that APL-16-5 gradually lost its activity of reducing the level of PA (Fig. 5i) and inducing PA ubiquitination in cells (Fig. 5j) and in cell-free assays (Fig. 5k) at high concentrations, which further supports a bifunctional mechanism. In contrast, APL-16-5 did not affect the degradation of SIRT7 by TRIM25 (Fig. 5l), which rules out the possible non-specific inhibition of TRIM25-mediated protein degradation at high concentrations of APL-16-5. Together, these data demonstrate that APL-16-5 acts as a PROTAC by binding to both TRIM25 and PA, and induces TRIM25-mediated ubiquitination and subsequent degradation of PA.”

We have investigated which motif of APL-16-5 binds to TRIM25 or PA by examining the three asperphenalenone compounds APL-16-5, APL-16-2, and APL-16-1. These three compounds are composed of a linear diterpene derivative linked to a phenalenone derivative via a C-C bond, which shares the identical phenalenone structural fragment with different diterpene structural fragment (especially the terminal part of the diterpene, seeing below). The results of BLI showed that all three compounds bound to PA with similar fitted K_D values, whereas APL-16-5 and APL-16-1, but not APL-16-2, interacted with TRIM25. This suggests that the phenalenone fragment, shared by all three compounds, mainly contributes to their interaction with PA, and the diterpene fragment determines the binding to TRIM25 (different structures with different binding

ability). We have now discussed the SAR analysis on page 17.

Referee #3

1. PROTACs are a hot-topic currently, and it's attractive to think that APL-16-5 functions by such a mechanism – indeed, the bivalent nature of the molecule even resembles a PROTAC. PROTACs typically work by inducing ternary complex formation between a target and an E3. While the authors do show weak binding to TRIM25 and PA by BLI (Figure 5 d through f), they fail to demonstrate interaction with the polymerase or ternary complex formation between the polymerase and TRIM25. The biophysical characterization (MS, purity, etc) for the recombinant proteins is also not provided.

Answer: We have performed different assays to demonstrate that APL-16-5 promotes the interaction between TRIM25 and PA, these include co-immunoprecipitation (Fig. 5a), BLI (Fig. 5h), and in situ proximity ligation assay (PLA) in living cells (PLA) (Fig. 5b). PA is a key subunit of influenza virus polymerase complex which consists of PA, PB1 and PB2. Depleting any of the three subunits abolishes viral RNA transcription and replication. Our data showed that APL-16-5 specifically targets PA instead of all three subunits (Fig. 2b), which suffices to block the formation of functional viral polymerase complex (Fig. 2a). To further demonstrate that APL-16-5 promotes the interaction of TRIM25 with PA but not with PB1 nor PB2, we performed the PLA experiments in cells expressing PA, PB1 and PB2, and observed interactions between TRIM25 and PA, not with PB1. We have now presented these data in the new Extended Data Fig. S5c, and described the results on page 13 as the following:

“Of note, results of PLA showed interactions of TRIM25 with PA (Extended Data Fig. S5c), but not with PB1 in cells expressing PA, PB1 and PB2, suggesting that PA but not the other subunits of viral polymerase complex is targeted by APL-16-5.”

In addition, we have now provided the information on the purity of the recombinant proteins in the Material and Method section in the revised manuscript.

2. It is common in the targeted protein degradation field to characterize PROTACs by global proteomics to obtain a more complete picture of what proteins are degraded. The authors should obtain these data through a suitable collaboration. If the compound is a promiscuous degrader of many proteins, that would bring into question what its functionally relevant target(s) is(are).

Answer: The same question was also asked by reviewer 2. We have investigated the protein profile in the cells treated with APL-16-5 using a multi-omics approach. Briefly, an LC-MS/MS-based label-free quantitative proteomics approach was used to analyze protein expression profile in control and TRIM25-knockout cells with or without APL-16-5 treatment. In

total, 6503 proteins were detected by proteomics analysis (a full list of identified proteins is available upon request). Only one cellular protein (FN1) were significantly downregulated by more than $\log_2(-1)$ fold with a p value < 0.05 by APL-16-5 in a TRIM25-dependent manner. We also performed transcriptomic study using an Illumina HiSeq X Ten platform to identify the genome-wide transcriptional changes associated with APL-16-5 treatment. These data together showed that only FN1 was reduced by more than 2-fold, and 12 more host proteins showed a moderate reduction ($-0.5 > \log_2 > -1$) in a TRIM25-dependent fashion at the post-transcriptional level. Therefore, the expression of very few host proteins was affected by APL-16-5, which demonstrates the very high selectivity of APL-16-5-mediated degradation. We have now presented these data in the Extended Data Table S4-7, and described the results on page 17 as the following:

“To further assess the selectivity of degradation by APL-16-5, we have investigated the protein profile in the cells treated with APL-16-5 using a multi-omics approach (Extended Data Table S4-6), and found that only the level of FN1 was reduced by more than 2-fold in a TRIM25-dependent fashion at the post-transcriptional level, and 12 more host proteins showed a moderate reduction ($-0.5 > \log_2 > -1$) (Extended Data Table S7), demonstrating a high selectivity of APL-16-5 action.”

3. The SAR description of what happens with the APL derivatives with respect to degradation, ubiquitination, and antiviral activity was confusing. It would be good to show the chemical structures of the compounds being described in the main figure and what each modification does to the different biological activities. It was not clear to me that the SAR data support the hypothesis that the compounds work as conventional PROTACs.

Answer: We have now presented the chemical structures of compounds APL-16-5, APL-16-2, and APL-16-1 in the revised Fig. 1, and added the SAR description on Page 17 in the revised manuscript. APL-16-5, APL-16-2 and APL-16-1 all have a linear diterpene derivative which is linked to a phenalenone derivative via a C-C bond. The phenalenone structural fragment is identical between these three compounds, whereas the diterpene structural fragment (especially the terminal part of the diterpene) differs. The results of BLI showed that all three compounds bound to PA with similar fitted K_D values, APL-16-5 and APL-16-1 but not APL-16-2 interacted with TRIM25 (Fig. 5d, 5e and Extended Data Fig. S5b and S5c). In agreement with the ability of binding to both TRIM25 and PA, compounds APL-16-5 and APL-16-1 but not APL-16-2 causes PA degradation and inhibits influenza virus replication.

Our initial analysis suggests that the identical phenalenone fragment, shared by all three compounds, may contribute to their interaction with PA, and the diterpene fragment, which

differs between the three compounds, may bind to TRIM25 with different affinities. The different binding affinity values to TRIM25 suggest that the hydroxyl groups, substituted to terminal part of the diterpene (C-29, C-30, and C-33 positions in the structure), may enhance the interaction with TRIM25, and the straight form of the diterpene moiety (influenced by the cis-trans configuration of the double bond) may increase the accessibility to TRIM25. The detailed SRA awaits further investigation with more compounds.

4. While APL-16-5 has antiviral activity and also reduces the half-life of PA in cellulo, there is somewhat of a disconnect between the concentrations at which these activities are observed. The antiviral EC₅₀ is 280 nM (Fig 1), but the half-life study and other experiments in Fig 2 monitoring depletion of PA were conducted at 2-10 micromolar, an order of magnitude higher. This raises the possibility that an additional target (or targets) contribute significantly to APL-16-5's antiviral activity. Again, this is an area where showing a correlation between SAR of the antiviral activity and the PA-depletion activity could be helpful.

Answer: We agree that, in addition to depleting viral PA protein, APL-16-5 may inhibit influenza virus by other mechanisms, given the lower antiviral EC₅₀ values compared to that of degrading PA. Along this line of possibility, we found that the drug-resistant viral mutant was still inhibited by high concentration of APL-16-5 albeit to a much less extent compared with the wild type virus (Fig. 3b and 3c), although the mutated PA was almost fully resistant to degradation by APL-16-5 (Fig. 3d). We have now discussed this further in the revised manuscript on page 9 as the following:

“We noted that the drug-resistant viral mutant was still inhibited by high concentration of APL-16-5 albeit to a much less extent (Fig. 3b and 3c), although the mutated PA was almost fully resistant to degradation by APL-16-5 (Fig. 3d), which suggests that APL-16-5 may act on more targets than PA to achieve its potent antiviral activity.”

In addition, we have measured the effect of another antiviral compound APL-16-1 on the stability of PA, and found that similar to APL-16-5, APL-16-1 was also able to reduce the half-life of PA protein from 18.8 h in control cells to 6.58 h in the treated cells (Fig. 2d). Moreover, increasing the expression of the PA protein gradually alleviated the inhibitory effect of APL-16-1 on IAV replication (Extended Data Fig. S3b). These new data provide further evidence supporting the correlation between SAR of the antiviral activity and the PA-depletion activity. We have now presented these data in the new Fig. 2d and Extended Data Fig. S3b, and described the results in the main text (page 7 and 9).

5. Does APL-16-5 exhibit the characteristic hook effect of PROTACs at higher concentrations? Showing this for antiviral activity and depletion of PA in cellulo would also bolster the mechanistic claims.

Answer: We have now tested the hook effect by performing a dose-response analysis of PA degradation and ubiquitination at high concentrations of APL-16-5. Indeed, increasing the concentration of APL-16-5 gradually diminished its activity of degrading PA (Fig. 5i) and inducing PA ubiquitination in cells (Fig. 5j) and in cell-free assays (Fig. 5k). In contrast, no effect was observed on the degradation of SIRT7 by TRIM25 (Fig. 5l), further supporting the specific inhibition of TRIM25-mediated degradation of PA at high concentrations of APL-16-5. The observed hook effect further supports that APL-16-5 acts as a PROTAC by binding to both TRIM25 and PA and inducing TRIM25-mediated ubiquitination and subsequent degradation of the PA protein. We have now presented these data in the new Fig. 5, and described the results on page 14 in the revised manuscript (Please also see our response to reviewer 2).

6. I'm not familiar with the TRIM25 ligase complex, but are there other components that can be shown to be essential through knock-down or other inhibitors (such as the Neddylation inhibitors used for Cul4 ligases) to further demonstrate the functional relevance of the complex in the inhibitors mechanism of action?

Answer: Thank you for this suggestion. Unfortunately, different from the cullin RING ligases that form multi-subunit complexes, the TRIM family proteins are 'single protein RING fingers', and alone can complete both substrate recruitment and catalysis activities. Little is known about how TRIM25-mediated K48-linked polyubiquitination is regulated and no specific TRIM25 inhibitors have been reported. We have conducted both knock-down and knock-out of TRIM25 which abolished the antiviral activity of APL-16-5 by preventing PA polyubiquitination and degradation (Fig. 3). These results support a key role of TRIM25 in APL-16-5-mediated degradation of PA.

7. The concentrations of APL-16-5 used in many of the experiments (Fig. 2d-2f, Fig 3, Fig 4, and most of the other figures in both the main paper and the extended data section) are not clearly indicated on the figure or in the figure legend. Since interpretation of the data requires this information, it should be made clear.

Answer: We have now added the compound concentrations in all the figure legends in the revised manuscript.

8. For the experiments with the resistant mutants in Fig 3, what was the concentration used? The authors should show titration curves and also provide some evidence as to the mechanism of resistance – e.g., reduced binding of APL-16-5? Increased stability of the mutant protein? The

mutations somehow affect accessibility of sites of ubiquitination – and show that this effect is sufficient for the restoration of viral replication

Without these data, many other potential mechanisms are possible, and the authors' claim that the compound is working as PROTAC isn't sufficiently supported. Overall, I think the data supports the conclusion that the compound has antiviral activity that is TRIM25 dependent but more work is needed to corroborate the details off the molecular mechanism.

Answer: The concentrations of the APL-16-5 are 2 and 10 μ M. The selected IAV showed a 7.5-fold increase in the EC₅₀ of APL-16-5 compared with that of the wild-type IAV, and remained fully susceptible to the inhibition by polymerase inhibitor ribavirin. We have performed the titration experiments and now presented the titration curves results in the new Extended Data Fig. S3d.

We have also investigated the mechanism of resistance. First, these mutations significantly increase the stability of PA with APL-16-5 treatment (Fig. 3d). Second, we have performed the co-immunoprecipitation experiment and observed the loss of TRIM25 interaction with the PA mutant in the presence of APL-16-5 (Fig. 3c) and the loss of APL-16-5-induced degradation of the PA mutant which is resistant to APL-16-5 (Fig. 3d). These data suggest that the resistance mutations prevent APL-16-5 from binding to PA. We have now presented these results in the new Extended Data Fig. S5b.

With these data and results of the other experiments including the demonstrated hook effect, it is convincing to conclude that the major antiviral mechanism of APL-16-5 is acting as PROTAC to target and degrade viral PA protein.

Fig 2B, reduction in PA is not concentration dependent, even at the two concentrations shown.

Also, there does appear to be a significant reduction in PB1.

Answer: We have now quantified the data of three independent Western blots and summarized the results in the new Extended Data Fig. S2a. The results showed that APL-16-5 caused a significant reduction in the level of PA, but no significant decrease in level of PB1, PB2, or NP, which supports the selective targeting of PA by APL-16-5. These data are now described on page 5 in the revised manuscript.

Reviewers Comments:

Referee #2 (Remarks to the Author):

the authors have satisfactorily answered reviewers' questions.

Referee #3 (Remarks to the Author):

This is an interesting study. I commend the authors on discovering a natural product with impressive anti-IAV activity in cell culture and in vivo models and on pursuing an exciting hypothesis that the compound's antiviral mechanism of action is to act as a PROTAC inducing TRIM25-mediated ubiquitination and degradation of PA. While the authors have done much work, their conclusion that APL-16-5 is still not adequately supported by data. What they have show is that APL-16-5 inhibits IAV and depletes PA via a TRIM25-dependent mechanism. The data do not demonstrate that APL-16-5 is a bonafide degrader. While the simple story is probably the most appealing, there are enough pieces of data to suggest that the mechanism and mode of action are not as simple as presented. Since the PROTAC mechanism is a major part of the significance of the work for the Nat Chem Biol audience (versus a more virological journal), it's critical to demonstrate the mechanism and to show that the PROTAC activity is the major main source of antiviral activity. Unfortunately, the paper currently falls short of this and therefore seems more appropriate for a specialized virology/antivirals journal.

Major issues:

1. Mismatch between antiviral data and degradation activity: 0.28 to 0.36 μ M EC50 with ~90% inhibition observed at 1 μ M (Fig 1b) but in Figure 5i, depletion of PA is NOT observed at 0.76 μ M APL-16-5. Appreciable depletion is not observed until 3-25 μ M, some 10-100-fold higher concentrations. If APL-16-5's antiviral activity is largely due to activity as a PROTAC, then one would expect to see depletion and antiviral activity at comparable concentrations. This suggests that another mechanism is responsible for the compound's antiviral activity.

Response: We agree that APL-16-5 may exert its antiviral activity by more than one mechanisms, especially in light of the much lower concentration of APL-16-5 required to inhibit IAV infection than degrading PA in the transient transfection experiment. To clarify this, we have now revised our conclusion and description in the abstract, introduction and result part, and acknowledged in the discussion on page 17, line 15, "It should be noted that APL-16-5 may exert its antiviral activity by more than one mechanisms, especially in light of the lower concentration of APL-16-5 required to inhibit IAV infection than degrading PA in the transient transfection experiment, which awaits further investigation."

Even so, we also propose that in the transient transfection experiment shown in Fig 5i, much higher PA protein might have been expressed (to facilitate detection by Western blot) than in IAV infected cells, thus much more APL-16-5 is required to elicit appreciable degradation of PA protein shown in the Western blot. Along this line, the concentrations of APL-16-5 to inhibit IAV infection and the concentrations of APL-16-5 to reduce PA levels in the transient transfection experiment may not be directly comparable. In addition, the level of viral replication may not be in proportion to the level of PA, in another word, the limited reduction in PA may significantly impair viral infectivity. Although these hypotheses awaits further investigation, the observations such as that increasing the expression of the PA protein alleviated the inhibitory effect of APL-16-5 on IAV replication (Fig. 3a and Extended Data Fig. S3a) suggest the important contribution of reduced PA expression by APL-16-5 to its anti-IAV activity.

2. If comparing with ribavirin, the mode of action does not appear entirely to be inhibition of viral RNA replication or transcription. The lack of Gluc expression in the IAV mini-genome replicon (Fig 1d) could mean inhibition of viral RNA replication (the stated conclusion) but could also reflect destabilization (or degradation) of the viral RNAs and/or impaired translation of the viral mRNA. Also, it should be noted that in the time of addition experiment (Fig 1c), ribavirin, the validated RNA replication inhibitor control, shows activity at the 2-4h window but not after 4h. This is clearly different from the kinetic profile of APL-16-5, which shows roughly equivalent activities in the 2-4 and 4-6h windows. Why would depletion of PA in the 4-6h window matter when RNA replication has already happened? This suggests that the mechanism and mode of action are more complicated than the model presented. This should all at least be acknowledged with appropriate changes to the text to acknowledge

Response: Agree. The data herein can not exclude the possibility that the lack of Gluc expression in the IAV mini-genome replicon could reflect destabilization of the viral RNAs. Therefore, we have now revised our conclusion on page 5, line 16, “These data suggest that APL-16-5 inhibits influenza virus infection by diminishing viral RNA levels.”

One possible explanation for the prolonged inhibitory effect is the multiple functions of PA. It has been reported that beside viral RNA synthesis, PA plays roles in general virus-host interaction and host protein synthesis shutoff (PMID: 25070354). Therefore, it is possible that depletion of PA may impair its other functions other than RNA replication. To clarify this, we acknowledge on page 4, line 25, “The relatively prolonged inhibition by APL-16-5 (up to 6 hours) compared with ribavirin (up to 4 hours) may result from the different antiviral mechanisms by these two drugs.”

3. The data in Fig. 3 showing that the PA N228K/A704H double mutant was not depleted in the presence of APL-16-5 support the hypothesis that PA is the likely viral target of APL-16-5 – however, it’s unclear what the mechanism of protection is, and this affects the strength of the interpretation. Do the mutations reduce binding to APL-16-5? Although Fig S5b shows less co-IP of

the double mutant PA, this could have been answered more definitively by measuring Kd values for recombinant PA with the mutations by BLI, thermal shift assays, etc. Some of the data suggest that the mutations affect PA stability independently of the interaction with APL-16-5. For example, the graph for Fig 3d is a bit misleading because the negative control is set to 100% in each case even though the blots show pretty clearly that the single mutants are expressed at lower levels even in the absence of APL-16-5, suggesting that the mutations individually affect PA stability/half-life. These caveats need to be acknowledged if not addressed experimentally.

Response: We agree that measuring the binding of PA mutants to APL-16-5 should provide direct evidence on the resistant mechanism. During the previous revision, we have made great efforts, but still failed to express and purify the double mutant PA. Alternatively, we examined the effect of the double mutation indirectly using a Co-IP assay. To further clarify this, we acknowledge on page 14, line 10, *“It should be noted that although APL-16-5 failed to promote the interaction of TRIM25 with the double PA mutant in a Co-IP assay (Extended Data Fig. S5b), providing indirect evidence on the action of APL-16-5 and the resistant mechanism, measuring the binding of PA mutants to APL-16-5 should provide more direct evidence.”*

Indeed, the single mutant N228K or A704H expressed at lower levels than that of wild type PA, which may be due to either protein stability as the reviewer suggested or ineffective expression. The reason to set negative control to 100% in each case is to show clearly the relative change in protein level with APL-16-5 treatment. We have now acknowledged the lower expression of PA single mutant on page 9, line 21, *“The single mutant N228K or A704H expressed at lower levels, and showed moderate resistance to degradation by APL-16-5.”*

4. With respect to the TRIM25-mediated depletion of PA in the presence of APL-16-5, is it possible that additional factors (viral or host) are needed to form a stable complex? These factors would presumably be present in the cell-based and lysate-based experiments but not in the BLI. While I think it goes beyond the scope of the current study to investigate this and to identify other factors that may contribute to the activity, I think it’s important to acknowledge this possibility and the pieces of data that are consistent with this versus with the proposed model.

Response: Agree. We have acknowledged this possibility on page 11, line 14, *“This essential role of TRIM25 does not preclude the involvement of other cellular factors in APL-16-5-induced PA degradation, which awaits further investigation.”*

5. Likewise, there is a basal level of interaction of TRIM25 and PA observable in the co-IP experiments in Fig 5a and S5b that is augmented in the presence of APL-16-5 but that does not appear to require APL-16-5. This observation and its mechanistic implications need to be acknowledged and discussed.

Response: Since we did not observe the interaction of TRIM25 and PA in the BLI assay, we suspect that the low basal level association of TRIM25 with PA in the co-IP experiment represents either a non-specific binding or an indirect association in other unknown complex. We have

acknowledged the basal level association of PA and TRIM25 on page 13, line 4, “we performed co-immunoprecipitation and noted a low basal level association of TRIM25 with PA, this association increased with increasing concentrations of APL-16-5 (Fig. 5a). The low level of PA found in the immunoprecipitate may represent either a non-specific binding or an indirect association in other complex.”

6. Given the disconnect between the antiviral potency and the Kd values of the compound for PA and TRIM25, Fig 5a: The authors show a 4-fold greater interaction of PA with TRIM25 in the presence of the proteasome inhibitor MG132 by proximity liqation assay and pulldown and from this conclude that APL-165-5 functions as a PROTAC to promote the interaction of PA with TRIM25, inducing ubiquitination and degradation of PA. The authors state on lines 262-263 “When MG132 was used to block the degradation of PA by APL-16-5, the interaction between TRIM25 and PA strengthened, as shown by more and brighter fluorescent dots.” This one interpretation, but another interpretation is simply that by inhibiting the proteasome, the abundance of PA in the cell increases and one detects “greater interaction” simply because there is more there to interact. I see no evidence in these experiments that the interaction is “strengthened” by the PROTAC.

Responses: Agree. We have now revised this description on page 13, line 10, “When MG132 was used to block the degradation of PA by APL-16-5, more and brighter fluorescent dots were detected (Fig. 5b, lower panel), accompanied by increased levels of PA (Extended Data Fig. S5a). Quantification of fluorescence revealed a four-fold increase in the colocalization between TRIM25 and PA as a result of MG132 treatment (Fig. 5c).”

7. Last but not least, there is considerable literature on the role of TRIM25 in the RIG-I-mediated antiviral response to influenza and other RNA viruses. In addition, Sawyer and colleagues have shown that TRIM25 blocks influenza virus RNA chain elongation via an RIG-I-mediated mechanism (Meyerson et al., 2017, Cell Host & Microbe 22, 627–638). The point is that influenza virus – TRIM25 biology is well-studied and not simple. This paper ignores all of this prior work and tells a story that, while attractive, requires more scrutiny and care in terms of its interpretations and conclusions.

Response: We have now cited literature on the role of TRIM25 in the RIG-I-mediated antiviral response to influenza and other RNA viruses in the revised manuscript. In addition, we have now discuss the work on TRIM25 and influenza virus on page 18, line 22, “Recently, TRIM25 has also been reported to inhibit IAV RNA synthesis by directly binding to viral ribonucleoproteins and block viral RNA elongation (PMID:29107643). It is worth investigating if APL-16-5 mediates the specific recognition viral PA protein by TRIM25, resulting in the increasing TRIM25 in the site of viral RNA synthesis, thus enhances TRIM25-mediated inhibition of IAV RNA synthesis and viral infection. This may act as another antiviral mechanism in addition to PA degradation.”

Other issues with data and interpretation:

1. Fig 2b. Authors claim that APL-16-5 specifically depletes PA instead of all three subunits,

however there is a clear decrease in PB1 at the 2 μ M concentration. This should be acknowledged and if half-life and polyubiquitination data are available for PB1, it would be informative (but not necessary) to include in the supporting information.

Response: We did not describe this change for PB1 at 2 μ M of APL-16-5 in the manuscript, because we did not observe this reduction in other repeats, and this change in this experiment was not detected at 10 μ M of APL-16-5.

2. Fig 3. Legend states that the cells were transfected with increasing concentrations of PA – authors should clarify if they did a protein transfection of recombinant PA or if they are transfecting in different amounts of a PA-expression plasmid.

Response: We have now explicitly stated this as, “...with increasing concentrations of PA plasmid DNA”

3. Although the authors performed -omics experiments to address the specificity of APL-16-5, they don't show depletion of PA (or viral RNA) in these experiments. One wants to see specific depletion of the viral target(s) over depletion of host factors in the presence of the compound.

Response: The omics experiments were designed to examine the possible effect of APL-16-5 on cellular gene expression at the RNA and protein levels. Viral RNA and proteins were not expressed in the cells examined, to avoid the complication caused by the effect of viral proteins on cellular gene expression.

Reviewer comments -

Reviewer #3 (Remarks to the Author):

This is an interesting story, but the data currently presented are insufficient to conclude that APL-16-5's antiviral activity is due to PROTAC activity. It's an attractive hypothesis, but the authors' data at are best consistent with their hypothesis but fail to prove it and also insufficient to conclude other mechanisms. Key experiments that are now standard in the targeted protein degradation field are missing. This was outlined in the review of the original submission and includes demonstration of ternary complex formation by PA, TRIM25, and APL-16-5; global proteomics to determine specificity/selectivity and to demonstrate that APL-16-5 is not a promiscuous degrader; SAR for degradation activity corresponds to SAR for antiviral activity.

The experiments to demonstrate interactions of APL-16-5 with PA and TRIM25 rely on co-IP or proximity ligation in overexpression systems. The equilibrium dissociation constants measured for the compound with purified PA and TRIM25 are in the double-digit micromolar, which is inconsistent with the concentrations at which antiviral activity is observed.

Many of the experiments are presented in isolation without taking into account the literature characterizing rich interactions of influenza virus with TRIM25, including the demonstration that TRIM25 blocks influenza virus RNA chain elongation via an RIG-I-mediated mechanism (Meyerson et al., 2017, *Cell Host & Microbe* 22, 627–638). This effect on RNA synthesis is the same mode of action proposed for APL-16-5. Interpretation of results involving knockout, RNAi, and overexpression of TRIM25 need to be interpreted in the context of this prior work.

As I wrote in my original review, without more data many other potential mechanisms are possible, and the authors' claim that the compound is working as PROTAC isn't sufficiently supported. Overall, I think the data supports the conclusion that the compound has antiviral activity that is TRIM25 dependent and that APL-16-5 at higher concentrations is associated with TRIM25-mediated depletion of PA, but more work is needed to corroborate the details off the molecular mechanism. The data do not demonstrate that APL-16-5 is a bonafide degrader. Despite the appeal of the story that APL-16-5 is a degrader/PROTAC, there are enough pieces of data to suggest that the mechanism and mode of action are not as simple as presented. This includes the mismatch between concentrations at which antiviral activity versus PA depletion are observed and the K_d values of the interaction of APL-16-5 with PA and TRIM25 as well as the other issues outlined above and in my reviews of the prior versions of the manuscript. Since the PROTAC mechanism is a major part of the significance of the work for the Nature Communications audience (versus a more virological journal), it's critical to demonstrate the mechanism and to show that the PROTAC activity is the major main source of antiviral activity. Unfortunately, the paper currently falls short of this and therefore seems more appropriate for a specialized virology/antivirals journal.

RESPONSE TO REVIEWER COMMENTS

Reviewer #3 (Remarks to the Author):

1. *This is an interesting story, but the data currently presented are insufficient to conclude that APL-16-5's antiviral activity is due to PROTAC activity. It's an attractive hypothesis, but the authors' data at are best consistent with their hypothesis but fail to prove it and also insufficient to conclude other mechanisms. Key experiments that are now standard in the targeted protein degradation field are missing. This was outlined in the review of the original submission and includes demonstration of ternary complex formation by PA, TRIM25, and APL-16-5; global proteomics to determine specificity/selectivity and to demonstrate that APL-16-5 is not a promiscuous degrader; SAR for degradation activity corresponds to SAR for antiviral activity.*

Response: We agree with these concerns, and have now focused on demonstrating the potent anti-IAV activity of APL-16-5 both in cultured cells and in mice, as well as APL-16-5-mediated degradation of viral PA protein by engaging TRIM25. Accordingly, we have revised the title, abstract and introduction by removing the description of PROTAC. In the discussion, we propose that the action of APL-16-5 is consistent with the mechanism of a PROTAC, and also acknowledge that more studies are warranted to conclude APL-16-5 being a PROTAC. We have performed proteomic analysis of APL-16-5-treated cells (Extended data Table S4 to S7), and found only one gene FN1 whose level was reduced by more than two-fold, which demonstrates a very high selectivity and specificity of APL-16-5-mediated protein degradation.

2. *The experiments to demonstrate interactions of APL-16-5 with PA and TRIM25 rely on co-IP or proximity ligation in overexpression systems. The equilibrium dissociation constants measured for the compound with purified PA and TRIM25 are in the double-digit micromolar, which is inconsistent with the concentrations at which antiviral activity is observed.*

Response: We acknowledge this difference between these effective concentrations of APL-16-5, and agree that more than one mechanisms likely underpin the potent anti-IAV activity of APL-16-5. This is now discussed on page 16, line 11, "We could not exclude the possibility that APL-16-5 may exert its antiviral activity by more than one mechanisms, especially in light of the much lower concentration of APL-16-5 required to inhibit IAV infection than degrading PA in the transient transfection experiment. This discrepancy can be attributed to the lower PA level in IAV-infected cells than that from the transfected plasmid DNA. Nonetheless, the observation that increasing the expression of the PA protein alleviated the inhibitory effect of APL-16-5 on IAV replication (Fig. 3a) supports the important contribution of reducing PA expression to the anti-IAV activity of APL-16-5."

3. *Many of the experiments are presented in isolation without taking into account the literature characterizing rich interactions of influenza virus with TRIM25, including the demonstration that TRIM25 blocks influenza virus RNA chain elongation via an RIG-I-mediated mechanism (Meyerson et al., 2017, Cell Host & Microbe 22, 627–638). This effect on RNA synthesis is the same mode of action proposed for APL-16-5. Interpretation of results involving knockout,*

RNAi, and overexpression of TRIM25 need to be interpreted in the context of this prior work.

Response: We have now discussed our findings in the context of the known anti-viral activities of TRIM5. Please see page 16, lines 26, “*TRIM25 regulates the RIG-I-mediated IFN pathway, leading to antiviral response to influenza and other RNA viruses. APL-16-5 does not appear to interfere with the cellular function of TRIM25, since APL-16-5 did not affect the IFN response in cells (Extended Data Fig. S7). In addition, it was well tolerated by mice, with no signs of toxicity up to 500 mg/kg, suggesting the low toxicity of APL-16-5 and its potential use in patients. A direct antiviral role of TRIM25 was recently reported in a study showing that TRIM25 inhibits IAV RNA synthesis by directly binding to viral ribonucleoproteins and blocking viral RNA elongation, and this antiviral function of TRIM25 is independent of its E3 ligase activity. Since the anti-IAV action of APL-16-5 requires the ubiquitin ligase function of TRIM25, APL-16-5 should not act by enhancing the binding of TRIM25 to IAV ribonucleoproteins. We have thus engineered a strategy of using small molecules to endow TRIM25 with a new antiviral ability of degrading key viral enzymes by exploiting its E3 ligase function.*”

4. As I wrote in my original review, without more data many other potential mechanisms are possible, and the authors’ claim that the compound is working as PROTAC isn’t sufficiently supported. Overall, I think the data supports the conclusion that the compound has antiviral activity that is TRIM25 dependent and that APL-16-5 at higher concentrations is associated with TRIM25-mediated depletion of PA, but more work is needed to corroborate the details off the molecular mechanism. The data do not demonstrate that APL-16-5 is a bonafide degrader. Despite the appeal of the story that APL-16-5 is a degrader/PROTAC, there are enough pieces of data to suggest that the mechanism and mode of action are not as simple as presented. This includes the mismatch between concentrations at which antiviral activity versus PA depletion are observed and the Kd values of the interaction of APL-16-5 with PA and TRIM25 as well as the other issues outlined above and in my reviews of the prior versions of the manuscript. Since the PROTAC mechanism is a major part of the significance of the work for the Nature Communications audience (versus a more virological journal), it’s critical to demonstrate the mechanism and to show that the PROTAC activity is the major main source of antiviral activity. Unfortunately, the paper currently falls short of this and therefore seems more appropriate for a specialized virology/antivirals journal.

Response: We agree that more conclusive data are needed to define APL-16-5 as a PROTAC, and our data do support this possibility. In addition to the potential PROTAC mechanism, our study presents a microbial metabolite that acts as a novel, potent anti-IAV inhibitor by engaging a cellular E3 ligase TRIM25 to degrade a key viral enzyme PA, which hopefully is able to appeal to the broad readers of Nature Communications.